# Model Debiasing by Learnable Data Augmentation

## Abstract

Deep Neural Networks are well known for efficiently fitting training data, yet experiencing poor generalization capabilities whenever some kind of bias dominates over the actual task labels, resulting in models learning "shortcuts". In essence, such models are often prone to learn spurious correlations between data and labels. In this work, we tackle the problem of learning from biased data in the very realistic *unsupervised* scenario, i.e., when the bias is unknown. This is a much harder task as compared to the supervised case, where auxiliary, bias-related annotations, can be exploited in the learning process. This paper proposes a novel 2-stage learning pipeline featuring a data augmentation strategy able to regularize the training. First, biased/unbiased samples are identified by training over-biased models. Second, such subdivision (typically noisy) is exploited within a data augmentation framework, properly combining the original samples while *learning* mixing parameters, which has a regularization effect. Experiments on synthetic and realistic biased datasets show state-of-the-art classification accuracy, outperforming competing methods, ultimately proving robust performance on both biased and unbiased examples. Notably, being our training method totally agnostic to the level of bias, it also positively affects performance for any, even apparently unbiased, dataset, thus improving the model generalization regardless of the level of bias (or its absence) in the data.

## 1 Introduction

The learning capabilities of deep neural networks are nowadays universally recognized, especially in classification tasks, where deep architectures are able to fit large amounts of data, reaching unprecedented performance. Yet, being the learning process purely data-driven, such models can also extremely prone to fail (often with high confidence) whenever test samples are drawn from a different distribution (or domain).

One of the reasons for poor generalization also lies in the possible biases present in the training samples, i.e., whenever significant portions of data show spurious correlations with class labels. This can lead the trained model to learn the so-called "shortcuts" to classify data, thus failing to generalize properly (Beery et al., 2018; Geirhos et al., 2020). For example, a duck can be classified as such by a biased model due to the presence of blue water in the surroundings, and not for the actual bird shape and appearance. Hence, the very same model will likely fail in case the input image depicts a duck located on land or, vice versa, another bird located on water may be mis-classified as a duck. This kind of "shortcuts" are usually learnt since a *vast majority* of the samples are characterized by a spurious bias (e.g., ducks in water), while only a few are unbiased (e.g., ducks on land).

There are several approaches to cope with this problem, depending on whether the bias is known or not. In the *supervised debiasing* settings, a model is optimized in presence of biased data assuming the ground-truth knowledge of the bias. Such additional auxiliary annotations can be used to drive model optimization towards a data representation invariant to the domain (or attribute), as in Alvi et al. (2018); Kim et al. (2019); Li & Vasconcelos (2019); Wang et al. (2019); Ragonesi et al. (2021); Sagawa et al. (2020); Clark et al. (2019); Arjovsky et al. (2019) (see Figure 1(a) for a visual explanation). However, the availability of such additional domain labels is mostly unrealistic in many practical scenarios, as it would require a great effort during data annotation, and in some cases it could even be impossible to obtain since biases could not

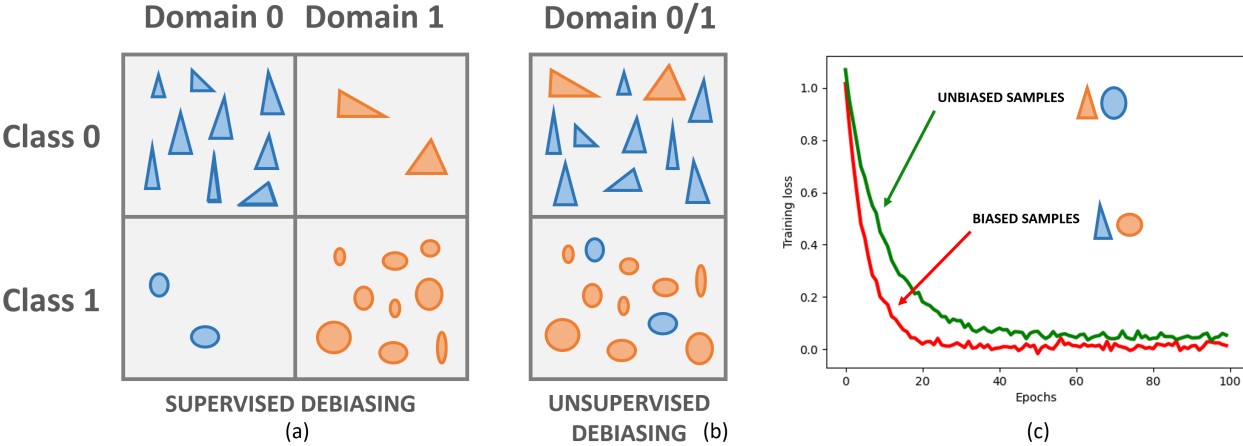

Figure 1: **Problem description.** (a) Biased datasets exhibit a class/domain imbalance, namely, one or more classes are mostly observed under one domain, leaving other options under-represented. In the case of supervised debiasing, additional information/annotations regarding the domain distribution are available. (b) In unsupervised debiasing settings, only class labels are available. Possibly, pseudo-labelling can be adopted to fall back into the supervised settings scenario. (c) Biased samples are fitted more easily than unbiased ones, as indicated by the different rates at which the average loss decreases. (Best viewed in color).

be directly interpretable, nor perceivable by humans. Hence, the need for methods that can achieve robust generalization in presence of bias, with no additional supervision nor labelling is strongly desirable.

The *unsupervised debiasing* scenario is in fact more realistic and challenging, and is the problem we are addressing here (Figure 1(b)). Assuming that the ground-truth annotations for the bias are not available, the idea is to attempt to (implicitly) infer such information and exploit it to debias our model. To this end, we design a data augmentation strategy, while properly weighting all the data samples proportionally to their bias level. Ultimately, the proposed approach aims at regularizing the training process by counterbalancing the biased samples with more "neutral", augmented examples, hence mitigating the learning of the bias, while achieving a successful generalization on the test set, often containing balanced (biased and unbiased) data.

In this paper, we devised a two-stage pipeline to tackle the unsupervised debiasing problem. First, we devise a methodology for the separation of biased/unbiased samples via a pseudo-labeling approach. Second, equipped with (noisy) bias/unbias pseudo-labels, we approach the problem of learning unbiased representations via a novel data augmentation strategy: we design a new objective function properly weighting the contribution of the two noisy subsets while learning an augmented version of the samples. Specifically, we generate (augment) new data by linearly interpolating biased and unbiased pseudo-labeled samples thus producing data which are more "neutral" than the original biased or unbiased data. This reduces the contribution of spurious correlations, which cause overfitting of biased samples as shown in Fig. 1(c). More specifically, we augment data by adopting a variant of Mixup (Zhang et al., 2018), i.e., a learnable version of it, where the parameters governing the data mixing are learnt, to increase the debiasing effect and boost the method's performance. It is important to note that, while existing methods (Nam et al., 2020; Clark et al., 2019) are primarily devoted to increasing accuracy on unbiased samples, overlooking the need for keeping high accuracy on biased data as well, we aim instead at achieving high accuracy over both types of data. Notably, the method is also completely agnostic of the presence of bias, as it shows positive effects even when no bias is (apparently) present in the data.

We test our method on typical datasets used in the literature, both containing synthetically generated and controlled bias (Corrupted CIFAR-10), as well as more realistic benchmarks (i.e., Waterbirds, CelebA and BAR), showing superior performance with respect to state-of-the-art methods.

In summary, the main contributions of our work are:

- We deal with the challenging *unsupervised debiasing* problem by proposing a novel two-stage approach to tackle it: first, we propose an initial data sorting technique aimed at segregating biased and unbiased samples; second, we employ a learnable data augmentation approach to properly regularize the training, limiting the detrimental effect of the bias.

- Our proposed approach learns unbiased models from biased and unbiased samples, by weighting their contributions through mixing. Specifically, we propose a principled data augmentation approach, consisting in learning the parameters controlling the mixing of the data samples. By considering augmented (mixed) data in the loss function, we inject a strong regularization signal in the training process, thus compensating the imbalance problem and leading to a more general and unbiased representation learning.

- We present an extensive validation of the proposed pipeline on datasets with controlled bias as well as realistic benchmarks, outperforming state-of-the-art approaches by a significant margin.

The rest of the paper is structured as follows. Section 2 discusses the related literature and highlights the original contributions of our proposed approach. Section 3 introduces the problem formulation and the related notation. The initial biased/unbiased data subdivision process is reported in Section 4. Section 5 instead details the 2nd stage consisting in the design of a loss function achieving invariant representation learning based on sample weighting and data augmentation via the learning of the mixing parameters to control the regularization. Sections 6 and 7 present the results and a thorough ablation analysis, respectively. Finally, Section 8 draws conclusions and sketches future research directions.

## 2    Related Works

The problem of learning from biased data has been explored in past years mostly in the *supervised* setting, i.e. when labels for the bias factor are available. Several methods tackled it seeking an invariant data representation to such a known factor. These approaches relied on adversarial learning (Lemoine et al., 2018; Alvi et al., 2018; Kim et al., 2019), variational inference (Louizos et al., 2016; Moyer et al., 2018; Creager et al., 2019), Information Theory (Ragonesi et al., 2021), re-sampling strategies (Li & Vasconcelos, 2019), or robust optimization (Sagawa et al., 2020). In this context, it is notable to quote Invariant Risk Minimization (Arjovsky et al., 2019), which seeks an optimal representation invariant across domains, and EnD (Tartaglione et al., 2021), which proposes to insert an information bottleneck module to disentangle useful information from the bias. Recently, causal inference strategies are also addressed to mitigate bias issues, specifically in the context of multimodal (vision-language) models (such as in VQA systems (Patil et al., 2023), and multi-label image classification (Liu et al., 2022).

A more interesting scenario is recently taking place, which is the *unsupervised* debiasing case, i.e., when the data bias is unknown, which is the case we are addressing in this work (Bahng et al., 2020; Levy et al., 2020; Nam et al., 2020; Liu et al., 2021). Several approaches have been lately proposed, the most straighforward consisting in, first, identifying the bias, and then apply supervised techniques to reduce its impact (e.g., (Sagawa et al., 2019; Ross et al., 2017; Cadene et al., 2019; Sagawa et al., 2020; Selvaraju et al., 2019; Kim et al., 2021; Liu et al., 2021; Sohoni et al., 2020)). Unfortunately, these approaches are typically leading to suboptimal performances since the bias is typically not easily identifiable, nor sometimes can be clearly associated to a specific attribute of the data. Hence, they likely fail since the debiasing is only based on an approximated, hence inaccurate, estimation of the bias information, which proved to degrade the method's accuracy.

A large part of these techniques are based on the initial estimation or the learning of the unknown bias, which is subsequently exploited for the actual debiasing stage in a more robust manner. For example, some methods leverage additional supporting models (e.g., (Lemoine et al., 2018; Teney et al., 2022; Nam et al., 2020; Li et al., 2022; Kim et al., 2022a; Liu et al., 2021; Kim et al., 2019)), typically in the form of adversarial networks (Jung et al., 2023) or ensembles of classifiers (Kim et al., 2022b), to learn the biases in the data, and use them to condition the primary model in order to mitigate their effects. For instance, (Kim et al., 2019)) tries to decouple the bias contribution in the feature embeddings by first minimizing the mutual

information between such embeddings and the bias itself. A debiased model is thus adversarially trained against such feature embedding network by leveraging an additional network to predict the bias distribution. At convergence, such bias prediction network is not able to predict the bias, while the feature embedding network has successfully "unlearnt" it. The method in (Jung et al., 2023) introduced a style-transfer GAN trained with several biased appearances. Target model bias-invariant representations are then obtained via contrastive learning between images affected by different biases. Other works (e.g., (Hwang et al., 2022; Nam et al., 2020; Seo et al., 2022; Ragonesi et al., 2023)) attempt to identify the so-called *bias-aligned* and *bias-conflicting* samples (another way to name *biased* and *unbiased* samples, respectively), followed by an adaptive re-weighting/re-sample procedure regularizing the training. The goal here is to debias the training process by properly weighting the individual samples on the basis of the (estimated) level of bias they are affected, so that bias-conflicting examples, usually very few, are generally given more importance with respect to the more numerous bias-aligned data. Several methods adopt this procedure leveraging a validation set to support the identification of the biased samples (Liu et al., 2021; Nam et al., 2022). For example, Nam et al. (2022) proposed a method called SSA (Spread Spurious Attribute) that follows a 2-stage strategy: the first stage consists in pseudo-labeling the samples as biased or unbiased with the help of a bias group-labeled set (training samples with bias attribute annotations); the second stage uses Group DRO (Sagawa et al., 2020) for the actual model debiasing, which also exploits such labeled validation set. Although this class of methods reached reasonable performances, the need of a group-annotated validation set makes it not fully unsupervised. contrast, our approach does not use any bias group-labeled annotation, while still being competitive in most of the tested datasets, when compared with such weakly bias-supervised methods.

Self-supervised or clustering mechanisms (Hong & Yang, 2021; Jung et al., 2021) are also used to promote the aggregation of samples with the same target class, but different bias. Generative bias-transformation networks are also used as translation models to transform the bias in the samples, so that bias-invariant representations can be learnt via contrastive learning. Similar "data transformation" approaches are also designed, aimed at leveraging the generation of new samples, often under the form of data augmentation, in order to counterbalance the effect of the majority of the biased examples during training. (Kim et al., 2021; Hwang et al., 2022).

Another class of methods tackles the debiasing task considering robust risk minimization (Bahng et al., 2020; Levy et al., 2020) in training, i.e., by learning a (debiased) network that is statistically independent from a model trained to be strongly biased by design. Such "bias amplification" methods, as (Lee et al., 2023), push to an extreme the training of a model to be strongly biased, that is, overfitted to the bias, to split between bias-aligned and conflicting examples. A different approach is proposed in (Shrestha et al., 2022), which operates at the architectural level, i.e., the network architecture is modified to impose inductive biases that make the network robust to data bias. The first inductive bias is imposed in order to use the most little network depth as needed for an individual example. The second inductive bias is instead pointed to use fewer image locations for prediction.

For completeness, even if these works are not dealing with the proposed approach, vision-language models are starting to be used to detect and mitigate biases in classification models, e.g., as in (Eyuboglu et al., 2022; Kim et al., 2024; Zhang et al., 2023).

To recap, it can be noted that the debiasing approaches in unsupervised scenarios share similar logical structures while differentiating from the specific methods to identify the bias attributes or estimating bias-aligned or -conflicting samples. Such methods span from the adoption of ensembling or ad-hoc auxiliary models, individual data re-weighting and re-sampling, clustering, the use of contrastive approaches or the design of special networks forcing feature orthogonality, to the generation of new unbiased data (augmentation) aimed at regularizing the training.

Differently from several former methods (Teney et al., 2022; Kim et al., 2022a; Nam et al., 2020; Bahng et al., 2020), our work does not rely on an ensemble of networks to have a reference biased model, neither we perform data upsampling as in (Liu et al., 2021), (Nam et al., 2020) and (Li & Vasconcelos, 2019), nor we consider the confidence of the predictions (as in (Kim et al., 2021)). Instead, we pursue an original pseudo-labeling approach to split the dataset in two subsets, biased and unbiased, which we subsequently use together with newly generated (augmented) data to regularize the overall training optimization process

and debias the model. More specifically, we adopt a data augmentation approach combining biased and unbiased samples: inspired by *mixup* (Zhang et al., 2018), we mix samples presenting peculiar features of the bias regime (likely representing a shortcut to infer the class) with samples that are deemed not being affected by it, so to produce new, more "neutral" examples able to counterbalance the bias influence. This is done by devising an adaptive adversarial mechanism that aims at maximizing the classification performance, while learning the parameters governing the mix of the samples. The adversarial mechanism is in fact introduced to challenge the classifier with augmented, difficult to correctly classify, samples, which are generated specifically to reduce the detrimental contribution of the bias during training. By optimizing such data augmentation mechanism by learning the best parameters of the mixing, we generate synthetic data expected to heavily reduce the spurious correlations that affect the original data, eventually leading to a stronger regularization and effectively debias the model during the training process.

## 3 Problem formulation

We consider supervised classification problems with a training set $\mathcal{D}_{train} = \{x_k, y_k, d_k\}_{k=1}^N$, where $x_k$ are raw input data (e.g., images), $y_k$ class labels, and $d_k$ domain labels. In the case of a biased dataset, $\mathcal{D}_{train}$ has several classes $y^i, i = 1, ..., C$, which are considered to be observed under different domains $d^j, j = 1, ..., D$. In general, $D$ can be different from $C$ but here, for clarity and without losing generality, we consider the case of $D = C$. When the majority of samples of a class $y^i$ is observed under a single domain $d^j$, we say that there is spurious correlation between $(y^i, d^j)$, i.e., the dataset has a bias.

We define $\mathcal{D}_{bias}$ as the subset of training samples that exhibit spurious correlations and $\mathcal{D}_{unbias}$ as the under-represented subset. Such subsets are highly imbalanced, i.e. $|\mathcal{D}_{bias}| >> |\mathcal{D}_{unbias}|$, which causes the problem of models learning "shortcuts". Conversely, when the bias does not account for the vast majority of class samples, its impact can be weaker, or even negligible, and the data would not be considered as affected so strongly by the bias anymore. Please note that the data from $\mathcal{D}_{bias}$ is not of lower quality but differs from $\mathcal{D}_{unbias}$ only in the frequency. While previous literature sometimes uses the notation *bias aligned* and *bias conflicting* samples (Nam et al., 2020) for $\mathcal{D}_{bias}$ and $\mathcal{D}_{unbias}$, we opt for a simpler terminology.

As an example, in a cats vs. dogs classification problem, most of the cats may be observed in an indoor home environment, while most of the dogs may be observed in outdoor scenes. For both classes, very few images are outside their main distribution, and models could identify the domain (indoor/outdoor) as the class.

When domain labels $d$ are not available, nor we have access to other bias information, we are facing the *unsupervised debiasing* problem. Hence, we consider a training set only containing input data and class label, $\mathcal{D} = \{\mathbf{x}, \mathbf{y}\} = \{x_k, y_k\}_{k=1}^N$. We consider the task of training a neural network $f_\theta$ on $\mathcal{D}$, with parameters $\theta$, which are usually found via Empirical Risk Minimization (ERM), i.e., by minimizing the expected Cross-Entropy loss over the training data:

$$\theta^* = \arg\min_\theta \frac{1}{N} \sum_{k=1}^N \mathcal{L}_{CE}(y_k, f_\theta(x_k)) \tag{1}$$

In such scenario, when trained via ERM, a model focuses mostly on the more numerous biased samples, underfitting the unbiased ones: this results in a biased model that uses spurious correlations as shortcuts to make inference, instead of correctly learning the class semantics. In general, $\mathcal{D}_{test}$ follows a data distribution different from $\mathcal{D}_{train}$: indeed, in the literature, the several datasets adopted for model debias can be of different types and composition. In other words, the biased samples may not be the majority in the test set, hence, it is important to learn a model that can be efficiently deployed to deal with *both* biased and unbiased samples. As also briefly mentioned above, please also note that when a class is not observed predominantly under a single domain/bias, the dataset could not be considered to be biased: in fact, an ERM model can learn features of a category from a multitude of possible domains, hence mitigating the learning of spurious correlations.

Finally, it is also important to note that the notion of *domain* might be very fuzzy, being possibly not interpretable or even unknown: domains could in fact lack a manifest semantic meaning. Moreover, there

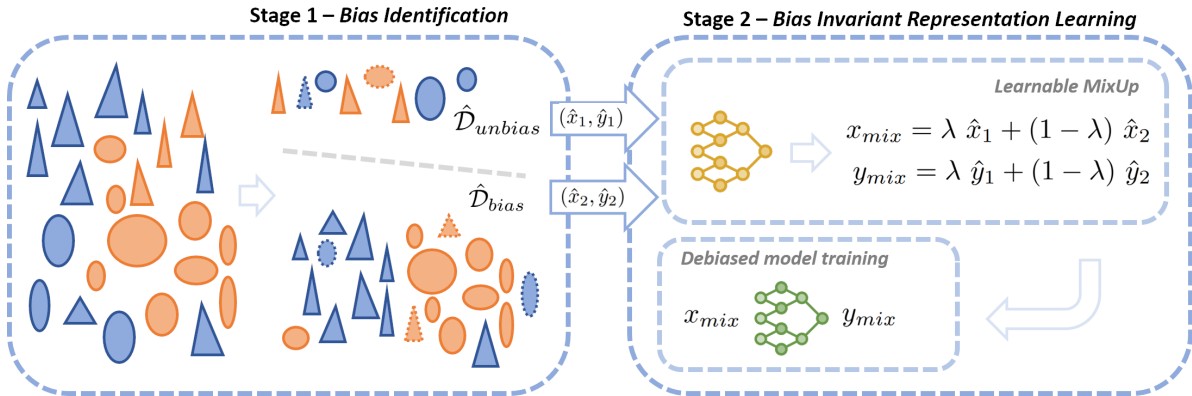

Figure 2: The proposed two-stage approach. In the first stage (*Bias Identification*), we estimate a separation of biased from unbiased samples through pseudo-labeling. In the second stage (*Bias Invariant Representation Learning*), we learn how to mix biased and unbiased samples (*Learnable MixUp*) in order to train a final debiased model (*Debiased model training*).

could also be the case of multiple predominant domains for a single class, and the opposite case of a same predominant domain shared among a subset of classes, or any scenario in-between. Typically, the more domains are involved, the less biased is the scenario, since heterogeneous training data may be sufficient to train a model with standard ERM without resorting to any special algorithm. To sum up, we are trying to face a realistic situation in which data samples are acquired without much control other than its class, resulting in a dataset potentially biased in an unknown manner or even unbiased in fortunate cases.

Our method tackles the unsupervised debiasing problem with a two-stage approach (Figure 2). In the first stage (Section 4), we try to separate biased from unbiased samples through a pseudo-labeling algorithm. Equipped with such (noisy) pseudo-labels, we train a model to produce a data representation that can accommodate both biased and unbiased samples regardless of the severity of the data bias (Section 5).

In the next section, in order to perform preliminary studies on a dataset with controlled bias, we use as a toy benchmark a modified version of the CIFAR-10 dataset, named Corrupted CIFAR-10, which is a modification of the original dataset (Krizhevsky et al., 2012a) that has been introduced in (Hendrycks & Dietterich, 2019). It contains 50,000 training RGB images and 10 classes, and each image is corrupted with a specific bias, represented here as a specific noise, e.g., Gaussian blur, salt and pepper noise, etc. Specifically, each class has a privileged type of noise under which it is observed during training (e.g., most of car images are corrupted with motion blur).

## 4 Bias Identification

In this first stage, our goal is to split the training set $\mathcal{D}$ into two disjoint subsets, $\hat{\mathcal{D}}_{bias}$ and $\hat{\mathcal{D}}_{unbias}$, which should reflect the actual ground-truth $\mathcal{D}_{bias}$ and $\mathcal{D}_{unbias}$.

In (Nam et al., 2020; Kim et al., 2021), it is shown how the biased samples are learnt faster than the unbiased ones: the imbalanced nature of the dataset makes the model more prone to learn first the numerous biased samples and later the unbiased ones. This behaviour can be observed by looking at the loss function trends of the two subsets (see Fig. 1(c)). As a proof of concept, we computed the Pearson Correlation Coefficient $\rho$ between correct/incorrect predictions and bias/unbias ground-truth labels for CIFAR-10 at different epochs of training of an ERM model (see Fig. 3, magenta plot). We observe that $\rho$ is sensibly higher than 0 as soon as the model fits the training data sufficiently (around $\rho = 0.6$ when the training accuracy reaches 75%). We computed the same metric for the loss value with similar results (blue plot). This shows that predictions of a vanilla ERM model are sufficiently correlated with ground truth bias/unbias labels, and therefore they can be exploited to divide the dataset.

We propose two approaches to assign to each samples the *bias/unbias* pseudo-label: the first one relies on predictions of an ERM model estimated at a specific epoch of the training process, where the epoch is determined fixing an achievable target training accuracy. The second approach instead takes into account the entire history of the predictions of a model along all of the training epochs. We call it identification "by single prediction" (SP) and "by prediction history" (PH), respectively.

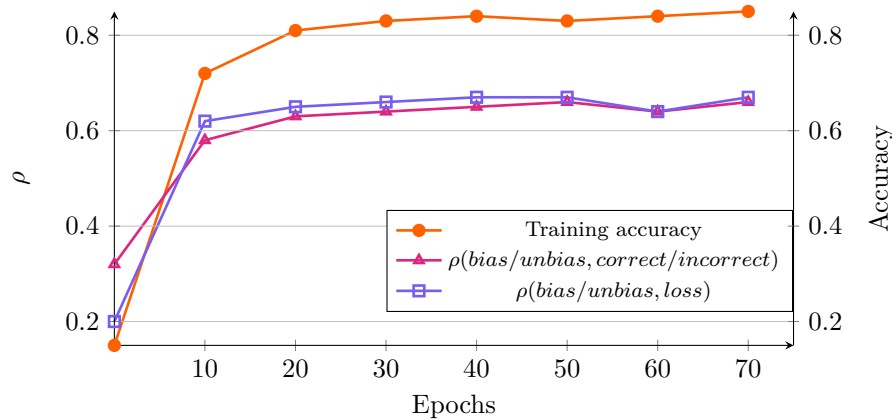

Figure 3: Training accuracy and Pearson correlation coefficient $\rho$ evolution over time. $\rho$ is computed between ground truth labels for $\mathcal{D}_{bias}$, $\mathcal{D}_{unbias}$ and the binary correct/incorrect label. Similarly we compute $\rho$ for bias/unbias and the loss values per sample.

## 4.1  Identification by Single Prediction

The SP stage consists in training a neural network $f_\phi$, with parameters $\phi$, via ERM until it reaches a target training accuracy of $\gamma$, where $\gamma$ is a hyper-parameter. This trained model is denoted as $f_\phi^\gamma$. When the model reaches the desired accuracy level, the training stops and a forward pass of the entire training set is performed: samples that are correctly predicted are assigned to $\hat{\mathcal{D}}_{bias}$ while those not correctly predicted are assigned to $\hat{\mathcal{D}}_{unbias}$. This should capture the fact that biased examples are fitted more quickly by the model. More formally:

$$
\begin{aligned}
\hat{\mathcal{D}}_{bias}^\gamma &= \{(x,y) \in \mathcal{D} \mid \sigma(f_\phi^\gamma(x)) = y\} \\
\hat{\mathcal{D}}_{unbias}^\gamma &= \{(x,y) \in \mathcal{D} \mid \sigma(f_\phi^\gamma(x)) \neq y\}
\end{aligned}
\tag{2}
$$

where $\sigma$ is the softmax function.

Using $\gamma$ as a hyper-parameter is convenient for two reasons. First, the setting of the amount of desired accuracy is dataset agnostic. This is different from prior work (Liu et al., 2021) that employs a similar strategy, but with the hyper-parameter controlling the number of epochs to train the model: in such a case, the number of epochs are strictly dependent on the specific dataset the model is trained on, and final results heavily depend on such parameter to be tuned accordingly. Second, we can easily have a precise control of the amount of samples assigned to the two splits. In real use cases, we do not know the correct assignments of the samples to the splits, and we have to rely on an a-priori setting of this parameter: we set $\gamma = 0.85$, which implies that 85% of training data are correctly classified and assigned to $\hat{\mathcal{D}}_{bias}$ while 15% are misclassified and assigned to $\hat{\mathcal{D}}_{unbias}$. 85% is a reasonable accuracy value, easily achievable by an ERM-trained model on a biased dataset. Besides, since in biased dataset a high percentages of samples are usually biased (e.g. 95% as minimum in former works (Nam et al., 2020; Kim et al., 2021)), setting $\gamma$ to high values is hardly a problem since the model overfits easily the training set given the multitude of biased samples. On the other hand, setting $\gamma$ too low implies that a lot of biased samples will be assigned to $\hat{\mathcal{D}}_{unbias}$, which is not desirable since will produce a very rough separation.

## 4.2 Identification by Prediction History

The PH method, instead of looking only at predictions at a specific epoch, considers the history of samples' predictions throughout the entire training stage. The rationale here is to make the splitting process more robust given that easy (biased) samples are often predicted correctly from the early training stages (often throughout the entire training), whereas hard (unbiased) samples could be difficult to be fit or are even left unlearned by the model. A second reason to prefer a multiple prediction approach is the fact that a prediction from a single epoch may be not representative of the actual difficulty of a specific sample: grouping predictions together is less affected by statistical oscillations.

Hence, the PH procedure consists in ERM training (as for the SP method), but at the end of each epoch, we perform a forward pass of the entire training set and check whether the prediction is correct or not, building a binary vector $\mathbf{s^t} = \{s_i^t\}_{i=1}^N$ of dimension $N \times 1$, that indicates what samples have been correctly/incorrectly $(0/1)$ classified at epoch $t$. At the end of training, we will have the $N \times K$ matrix $\mathbf{S}$, with $K$ total number of training epochs, which describes the history of predictions for each sample ( Figure 4).

Summing up along the epoch axis produces a new $N \times 1$ ranking vector $\hat{\mathbf{s}} = \{\hat{s}_i\}_{i=1}^N = \sum_{t=1}^K \mathbf{s^t}$ that describes how many epochs each sample has been correctly classified (see Figure 4, left). We then compute the histogram for $\hat{\mathbf{s}}$ (using $K$ bins - in order to span all possible values of the entries of $\hat{\mathbf{s}}$). We can observe that most of the samples are correctly predicted throughout the entire training ($K$-th rightmost bin in Fig. 4), whereas few samples are never or almost never correctly predicted (leftmost bin in Fig. 4, right). Empirically, we found that most of the samples in the leftmost bins are unbiased: rovided with this insight,

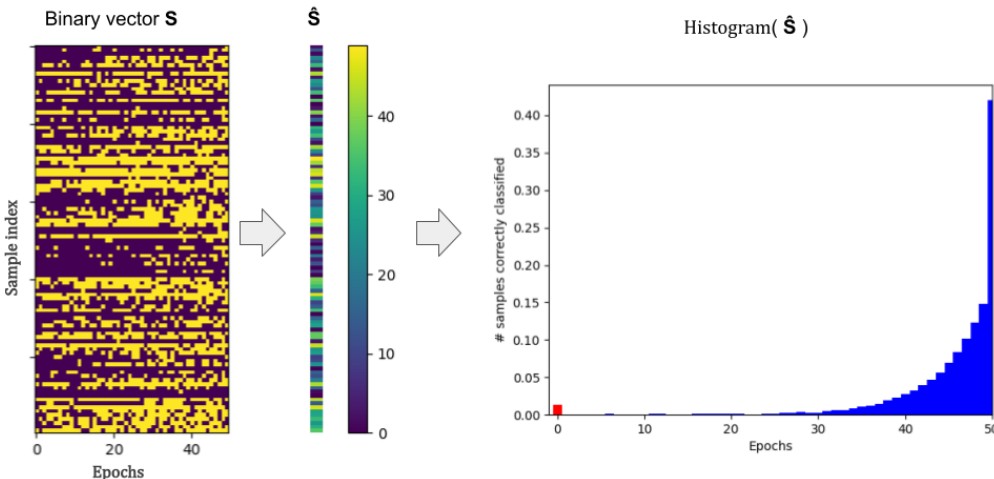

Figure 4: Binary matrix $\mathbf{S}$ on the left: rows represent samples indexes $i$ while columns represent epochs $t$. $s_i^t$ is 0 when sample $i$ is misclassified at epoch $t$ or 1 when it is correctly classified. When summing up along the epochs, we get the vector $\hat{\mathbf{s}} = \sum_{t=1}^K \mathbf{s^t}$ (middle) that represent the "prediction's history" of each sample after $K$ epochs. When computing its histogram (right), we observe that the distribution is severely skewed towards easy samples (those correctly predicted most of the epochs). We highlighted in red the leftmost bin which is significantly higher than its neighbors: this shows how samples that are never correctly classified are more than those that are rarely correctly classified.

we devised a strategy to iteratively split the dataset by identifying the samples to be assigned to $\hat{\mathcal{D}}_{unbias}$ and those to be included in $\hat{\mathcal{D}}_{bias}$.

To this end, we train an ERM model with cross-entropy, where each sample is weighted by $\mathbf{c} = \{c_i\}_{i=1}^N \in [0, 1]$. Values equal or close to 1 keep the learning rate unaltered or almost unaltered, whereas smaller values impact the learning rate by slowing it down. The vector $\mathbf{c}$ is initialized with all entries equal to 1; during training, we leave unaltered the weights for samples that are already well predicted while we decrease them for samples

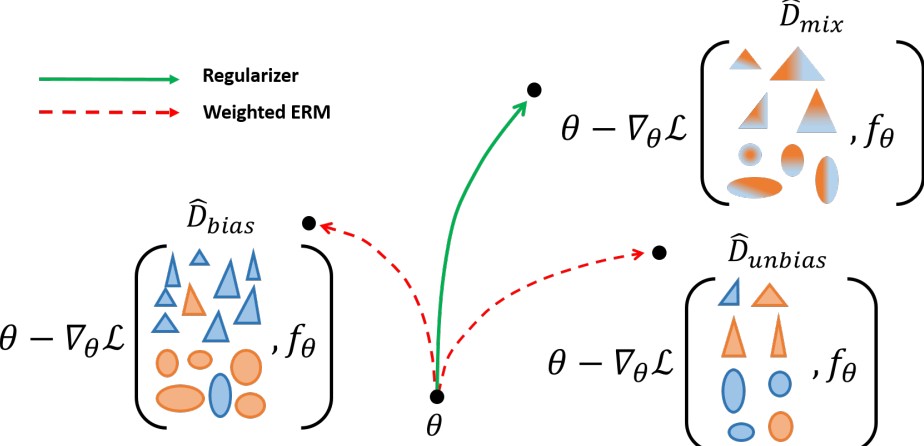

Figure 5: Gradients for $\mathcal{L}(\hat{\mathcal{D}}_{bias}, f_\theta)$ and $\mathcal{L}(\hat{\mathcal{D}}_{unbias}, f_\theta)$ are evaluated to produce the weighted ERM contribution. The regularization step using mixed data aims at producing an additional gradient contribution that decreases the loss function for $\hat{\mathcal{D}}_{mix}$ as for Equation 6. (Best viewed in color)

that are difficult to predict, thus implicitly decreasing their learning rate. Every $M$ epochs we compute the ranking vector $\hat{\mathbf{s}}$, and use it to modulate the weights vector $\mathbf{c}$ as follows:

$$\mathbf{c}^{new} = \mathbf{c}^{old} - (1 - \frac{\hat{\mathbf{s}}+1}{M}), \quad \hat{\mathbf{s}} = \sum_{t=1}^{M} \mathbf{s^t} \tag{3}$$

The value of $c_i$ is clipped in the range $[0, 1]$, so that the learning rate can only be decreased or left unaltered. Samples that are inherently difficult to be learned (i.e. samples for which $s_i^t$ is often zero) will have $\hat{s}_i \ll M$ and thus a low updated weight $c_i^{new}$. In the end, we are training a *bias-amplified* model.

At the end of training, after re-computing the histogram of the ranking vector $\hat{\mathbf{s}}$, we select all samples in the most difficult, leftmost, bin (those that have never been correctly predicted - see Fig. 4, right), and assign them to $\hat{\mathcal{D}}_{unbias}$, while assigning the others to $\hat{\mathcal{D}}_{bias}$.

## 5 Bias-invariant representation learning

Provided with pseudo-labels for the two estimated subsets $\hat{\mathcal{D}}_{bias}$ and $\hat{\mathcal{D}}_{unbias}$, we now have to deal with the problem of learning data representations that are not only suitable for the biased data, but can generalize well to unbiased samples too. The driving idea is to exploit the two resulting subsets, even if noisy, to generate augmented data in order to counterbalance the harmful effect of the numerous biased examples in training. Hence, we propose to combine biased and unbiased samples as an effective strategy to produce synthetic images that exhibit attributes from both $\hat{\mathcal{D}}_{bias}$ and $\hat{\mathcal{D}}_{unbias}$. Data augmentation is useful to attain oversampling of unbiased samples, which is a strategy sometimes used in bias removal methods. We follow Mixup (Zhang et al., 2018) as a state-of-the-art augmentation tool, which allows us to incorporate unbiased data and produce augmentations which are more effective for model debiasing.

As a preliminary analysis, we show in Sec. 5.1 how a convex combination (Zhang et al., 2018) of biased and unbiased samples results in a strong regularization that helps preventing the model from overfitting the biased data, when added to standard weighted average of the loss terms for the two noisy subsets (Ragonesi et al., 2023). We will refer to this baseline as *vanilla mixup*. Motivated by the fact that the mixing coefficients have a strong impact on the final accuracy, we thus explore in Section 5.2 how the strategy for mixing the samples can be *learned* directly from the data itself, in order to bring further improvement, leading to the proposed *learnable mix* or *l-mix* in short.

### 5.1 Vanilla Mixup as a bias regularizer

Given the two estimated pseudo-labeled subsets $\hat{\mathcal{D}}_{bias}$ and $\hat{\mathcal{D}}_{unbias}$, we consider a neural network $f_\theta$, with parameters $\theta$, trained from scratch.

**Weighted ERM.** In this step, we seek the best parameters $\theta$ on the basis of the two subsets $\hat{\mathcal{D}}_{bias}$ and $\hat{\mathcal{D}}_{unbias}$ via gradient descent. The $\theta$'s updating rule is:

$$\theta^* = \theta - \eta \; \nabla_\theta \; [(1 - \gamma) \; \mathcal{L}(\hat{\mathcal{D}}_{bias}, f_\theta) + \gamma \; \mathcal{L}(\hat{\mathcal{D}}_{unbias}, f_\theta)] \tag{4}$$

where $\eta$ is the learning rate. Note that in case the two subsets are obtained by *single prediction*, $\gamma$ is the same hyper-parameter introduced in the Sect. 4.1, i.e., the target training accuracy. In this step, we scale the two loss functions with $\gamma$ and $1 - \gamma$ to deal with data imbalance ($|\hat{\mathcal{D}}_{bias}| >> |\hat{\mathcal{D}}_{unbias}|$). To rebalance the contributions from the two splits, an obvious choice is to set the weights inversely proportional to the cardinality of the two subsets, which is nothing else than the fixed and controllable hyper-parameter $\gamma$ used to estimate the biased and unbiased subgroups. As discussed in the previous section, this value was fixed to $\gamma = 0.85$ and maintained on all experiments for all datasets. Similar values for $\gamma$ have not been shown to provide significant discrepancies in the final performance.

**Mixup as a regularizer.** Subsequently, we seek a representation that can conciliate both biased and unbiased samples and at the same time can prevent the model from overfitting the training data (especially $\hat{\mathcal{D}}_{bias}$). We take inspiration from Mixup (Zhang et al., 2018) as a way to combine samples from the two subsets. Mixup consists in using a convex combination of both input samples and labels demonstrating its efficacy as an effective regularizer. Specifically, we feed the model with synthetic (augmented) samples resulting from a *mix of examples from biased and unbiased subsets*, aiming at breaking the shortcuts present in the data (see Fig. 5).

We construct $\hat{\mathcal{D}}_{mix} = \{x_{mix}, y_{mix}\}$ by mixing examples of $\hat{\mathcal{D}}_{bias}$ and $\hat{\mathcal{D}}_{unbias}$ with corresponding labels $y$, as in the Mixup method (Zhang et al., 2018), by sampling the mixing parameter $\lambda$ from a Beta distribution, i.e.:

$$\begin{aligned}
(\hat{x}_1, \hat{y}_1) &\in \hat{\mathcal{D}}_{bias} \;, (\hat{x}_2, \hat{y}_2) \in \hat{\mathcal{D}}_{unbias} \\
x_{mix} &= \lambda \; \hat{x}_1 + (1 - \lambda) \; \hat{x}_2 \\
y_{mix} &= \lambda \; \hat{y}_1 + (1 - \lambda) \; \hat{y}_2 \\
with \quad &\lambda \sim Beta(\alpha, \beta)
\end{aligned} \tag{5}$$

Once the augmented samples $(x_{mix}, y_{mix})$ are computed, the model is updated combining the weighted ERM and the regularization term:

$$\mathcal{L}_{vanilla} := \underbrace{(1 - \gamma) \; \mathcal{L}(\hat{\mathcal{D}}_{bias}, f_\theta) + \gamma \; \mathcal{L}(\hat{\mathcal{D}}_{unbias}, f_\theta)}_{\text{Weighted ERM}} + \zeta \; \underbrace{\mathcal{L}(\hat{\mathcal{D}}_{mix}, f_\theta)}_{\text{Regularizer}} \tag{6}$$

where $\zeta$ is a hyper-parameter weighting the regularization term. The hyper-parameter $\zeta$ controls the amount of regularization in the final loss: if $\zeta = 0$, the method corresponds to standard (weighted) ERM in which the contributions of the losses on the two subsets are scaled by $(1 - \gamma)$ and $\gamma$. When $\zeta > 0$ the weighted ERM optimization trajectory is corrected by the regularization term (see Fig. 5 for a pictorial representation of the procedure). This corresponds to find parameters $\theta$ that are good for both $\hat{\mathcal{D}}_{bias}$ and $\hat{\mathcal{D}}_{unbias}$, but can also eventually reduce the loss value on the newly generated data samples $\hat{\mathcal{D}}_{mix}$. We will actually show that performance is not much affected by the choice of $\zeta$: as described in Sect. 7.4, accuracy increases and reaches a plateau (please, note that $\zeta$ axis is in log scale), and only with higher orders of magnitude (in the order of $\zeta \sim 10^3$) the regularization effect becomes detrimental. We set $\zeta = 10$ for all other experiments.

To conclude this preliminary analysis, we explored how the choice of the two parameters $(\alpha, \beta)$ impacts the final test accuracy on a standard dataset, CIFAR-10. As shown in Fig. 6, the convex combination of biased

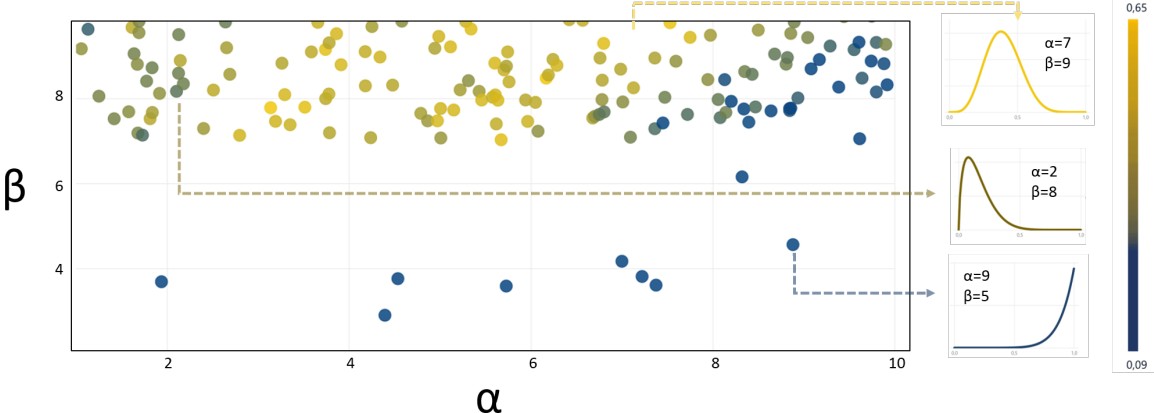

Figure 6: Performance of a model trained with augmented samples generated by Mixup for different pairs of parameters $(\alpha, \beta)$ (equations 5 and 6). Each dot represents a single run done with parameters of corresponding coordinates. The right color bar represents the model's test accuracy. We show 3 examples for different probability density functions corresponding to specific pairs $(\alpha, \beta)$.

and unbiased samples works reasonably well for a wide range of the parameters $\alpha$ and $\beta$ controlling the probability density function of the Beta distribution. We employed Bayesian optimization to explore the parameters space in order to highlight the region in which the generated samples gives the best accuracy on the test set. Please, note that in this preliminary investigation only, we used the ground truth annotations for $\mathcal{D}_{bias}$ and $\mathcal{D}_{unbias}$ in order to decouple the learning problem from the bias-identification problem). Figure 6 shows that the best combination is attained when the Beta distribution is slightly skewed towards the unbiased samples, namely, they are given more importance than the biased ones. Conversely, the worst scenario occurs when the distribution puts more emphasis to the biased samples. However, we claim that augmented data can be generated in a more principled way, that is, by *learning* $\alpha$ and $\beta$, as detailed in the next section.

## 5.2 Unbiasing by learnable data augmentation

Building on the insights gained from the analysis of Sec. 5.1, we designed an end-to-end pipeline that is able to simultaneously learn the classification task and optimize for the best mixing strategy. Our model (see Fig. 7) is composed of 1) a backbone network acting as feature extractor $f_\theta$, and 2) a classifier head $g_\phi$, with parameters $\theta$ and $\phi$, respectively, the latter producing logits from the features $f_\theta(x)$. We also adopt 3) a module $h_\psi$ which outputs two scalar values, namely, the parameters $\alpha$ and $\beta$ of the Beta distribution. Samples from $\hat{\mathcal{D}}_{bias}$ and $\hat{\mathcal{D}}_{unbias}$ are fed to $f_\theta$ and the corresponding feature vectors $f_\theta(\hat{x}_1)$ and $f_\theta(\hat{x}_2)$ are then concatenated. The module $h_\psi$ takes as input such concatenation, denoted as $\parallel$ in the equation below, and estimates the parameters $(\alpha, \beta)$ of the Beta distribution, which is subsequently sampled to extract the mixing coefficient $\lambda$:

$$(\alpha, \beta) = h_\psi(\, f_\theta(\hat{x}_1) \parallel f_\theta(\hat{x}_2)\,) \tag{7}$$

Specifically, we sample a vector of mixing coefficients $\lambda$'s (one for each pair in the batch) from the Beta distribution. We rely on the reparametrization trick (Kingma & Welling, 2014) in order to have a fully differentiable pipeline, since we need to differentiate through the sampling procedure. To do so, we employed the Pytorch implementation of the reparametrized Beta distribution. More formally, we sample $\lambda_i$ for the $i - th$ sample from the Beta distribution parametrized by $\alpha_i^*$ and $\beta_i^*$:

$$\lambda_i \sim Beta(\alpha_i^*, \beta_i^*). \tag{8}$$

Provided with $\boldsymbol{\lambda} = \{\lambda_i\}_{i=0}^N$, we compute $(x_{mix}, y_{mix})$ (Eq. 5), and use it as input for $f_\theta$, whose outcome is subsequently fed to $g_\phi$. The logits are then passed to the $g_\phi$ softmax layer $\sigma$ to infer $\tilde{y}_{mix}$, which is used as

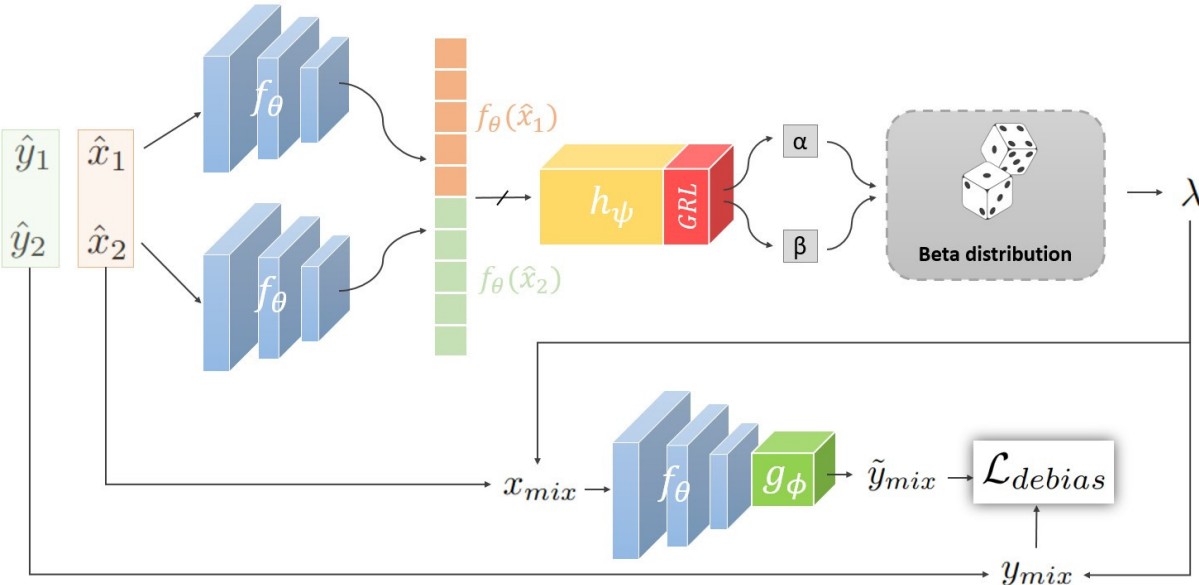

Figure 7: Proposed pipeline for learning the parameters of the Beta distribution. Features from $(\hat{x}_1, \hat{y}_1) \in \hat{\mathcal{D}}_{bias}$ and $(\hat{x}_2, \hat{y}_2) \in \hat{\mathcal{D}}_{unbias}$ are extracted with $f_\theta$, concatenated and fed to the module $h_\psi$, which predicts the $\alpha$ and $\beta$ that parameterize a Beta distribution, which, in turn, is sampled to obtain the mixing coefficient $\lambda$. The GRL layer on top of $h_\psi$ model ensures that when gradients are back-propagated, $\psi$ parameters are optimized to *maximize* the cross entropy. Namely, we want to produce mixed samples that are challenging for the classification network $g_\phi(f_\theta)$. Note also that gradients are stopped before going back to the initial encoders ($\nrightarrow$ denotes a StopGradient operation).

input for the Cross Entropy (CE) loss $\mathcal{L}_{CE}$.

Following the paradigm of adversarial training, we want to learn combinations of $x_1$, $x_2$ that generate challenging samples for the network, acting indeed as adversaries, i.e., for which the loss value *increases*. Such augmented samples are then used as training samples for effectively minimizing the loss function.

When doing backpropagation we want to seek the parameters $\theta$ and $\phi$ that minimize the CE loss, while at the same time looking for the combination – i.e., the parameter $\lambda$ drawn from the Beta distribution governed by parameters $\alpha$ and $\beta$ – of the two inputs $\hat{x}_1$ and $\hat{x}_2$ that maximize the CE loss. To do so, we apply a Gradient Reversal Layer (GRL) (Ganin & Lempitsky, 2015) to change the sign of the gradient when updating the weights $\psi$ (see Fig. 7).

Formally, the optimization problem we want to solve is:

$$
\begin{aligned}
\theta^*, \phi^* &= \underset{\theta,\phi}{\mathrm{argmin}} \ \mathcal{L}_{CE}(\sigma(g_\phi(f_\theta(x_{mix})), y_{mix}) \\
\psi^* &= \underset{\psi}{\mathrm{argmax}} \ \mathcal{L}_{CE}(\sigma(g_\phi(f_\theta(x_{mix})), y_{mix})
\end{aligned}
\tag{9}
$$

where, in the second equation, the dependency on $\psi$ is implicit in the variables $(x_{mix}, y_{mix})$. Specifically, this dependency is defined in equation 5, given that the mixing coefficients $\lambda_i$ are sampled from $Beta(\alpha_i, \beta_i)$, where $\alpha$ and $\beta$ depend in turn upon $\psi$, as per equation 7. Note that we prevent the gradient to flow into $f_\theta$

for the two streams by introducing a StopGradient operation (see Fig. 7, top left). Therefore, $f_\theta$ is updated only through $\frac{\partial f_\theta(x_{mix})}{\partial \theta}$ (see Fig. 7, bottom).

---

**Algorithm 1** Pseudocode for a single training iteration.
---

1: **function** TRAINING_ITERATION($\hat{\mathcal{D}}_{bias}, \hat{\mathcal{D}}_{unbias}, \eta$)
2:
3:      sample batches $(\hat{x}_1, \hat{y}_1), (\hat{x}_2, \hat{y}_2)$ from $\hat{\mathcal{D}}_{bias}, \hat{\mathcal{D}}_{unbias}$
4:      compute $f_\theta(\hat{x}_1), f_\theta(\hat{x}_2)$
5:      detach gradient from $f_\theta(\hat{x}_1), f_\theta(\hat{x}_2)$
6:      $(\alpha, \beta) = h_\psi(f_\theta(\hat{x}_1) \parallel f_\theta(\hat{x}_2))$
7:
8:      $\lambda \leftarrow$ sample from Beta($\alpha, \beta$)
9:      $x_{mix} = \lambda \hat{x}_1 + (1 - \lambda) \hat{x}_2$
10:     $y_{mix} = \lambda \hat{y}_1 + (1 - \lambda) \hat{y}_2$
11:
12:     $\tilde{y}_{mix} \leftarrow g_\phi(f_\theta(x_{mix}))$
13:     $\mathcal{L}_{debias} \leftarrow \mathcal{L}_{CE}(\tilde{y}_{mix}, y_{mix}) + \omega Reg$
14:
15:     # update networks' parameters
16:     $\theta \leftarrow \theta - \eta \nabla_\theta \mathcal{L}_{debias}$
17:     $\phi \leftarrow \phi - \eta \nabla_\phi \mathcal{L}_{debias}$
18:     $\psi \leftarrow \psi + \eta \nabla_\psi \mathcal{L}_{debias}$                          ▷ GRL inverts gradient sign
19: **end function**

---

**Regularizer.** We may optionally introduce a regularizer that acts as a prior knowledge on the objective function to lead to skewed distributions. Since we deal with imbalanced data, i.e. $|\mathcal{D}_{bias}| >> |\mathcal{D}_{unbias}|$, it is usually beneficial (see Fig. 6) to constrain $h_\psi$ to learn a family of Beta distributions that are skewed towards the minority group, i.e., the unbiased subset. This can be easily achieved by adding a regularization term $Reg$ that forces the expected value of the Beta distribution to be close to the (noisy) bias ratio. We chose the simple Mean Square Error as regularizer:

$$Reg = MSE(\ \frac{\alpha}{\alpha + \beta},\ \frac{|\hat{\mathcal{D}}_{unbias}|}{|\hat{\mathcal{D}}_{bias}| + |\hat{\mathcal{D}}_{unbias}|}\ ). \tag{10}$$

Thus, the total loss becomes:

$$\mathcal{L}_{debias} = \mathcal{L}_{CE}(\sigma(g_\phi(f_\theta(x_{mix})), y_{mix}) + \ \omega\ Reg, \tag{11}$$

where $\omega$ is a hyper-parameter that controls the amount of the regularization.

The several stages composing the proposed method and all introduced hyper-parameters will be analyzed in our ablation study in Sect. 7. The pseudocode for the (single iteration) training process is reported in Algorithm 1.

## 6   Experiments

We tested our proposed method on several benchmarks: a dataset with controlled synthetic biases, and different realistic datasets targeting classification tasks. We made a comparison with other existing methods tackling the debiasing problem in both supervised and, more extensively, unsupervised ways.

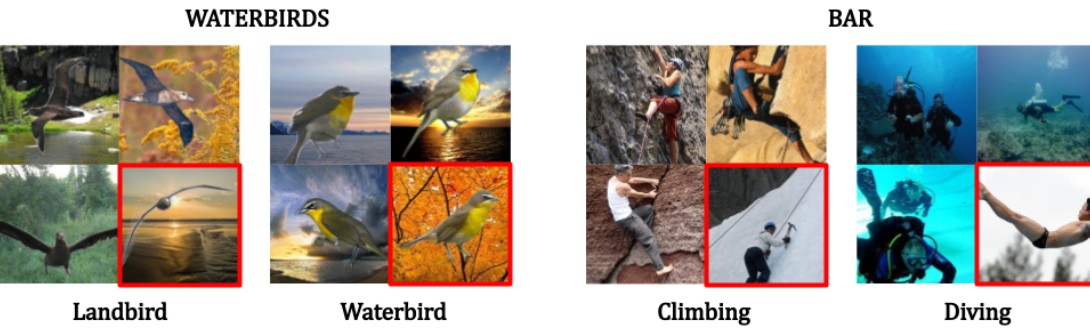

Figure 8:  Examples of training biased and unbiased (with red boundary) data, from Waterbirds and BAR.

## 6.1  Benchmarks

**Synthetic bias.** We consider a benchmark that has been introduced in Hendrycks & Dietterich (2019), namely corrupted CIFAR-10 in which the amount of bias is synthetically injected and thus controlled. Corrupted CIFAR-10 is a modification of the original dataset (Krizhevsky et al., 2012a) and has been adopted by Nam et al. (2020) in the context of debiasing. It consists of 50,000 training RGB images and 10 classes. The bias stems from the fact that each image is corrupted with a specific noise/effect (e.g., motion blur, shot noise, contrast enhancement, etc.). In practice, each class has a privileged type of noise under which it is mostly observed in training (e.g., most of images depicting trucks are corrupted with Gaussian blur).

**Realistic bias.** We prove the efficacy of our method on realistic image datasets, namely, Waterbirds, CelebA, and Bias Action Recognition (BAR). Waterbirds has been introduced in (Sagawa et al., 2020) and combines bird photos from the Caltech-UCSD Birds-200-2011 (CUB) dataset (Welinder et al., 2010), where the background is replaced with images from the Places dataset (Zhou et al., 2018). It consists of 4,795 training images and the goal is to distinguish two classes, namely *landbird* and *waterbird*. The bias is represented by the background of the images: most landbirds are observed on a land background while most waterbirds are observed in a marine environment (Fig. 8 - left).

CelebA (Liu et al., 2015) consists in over 200,000 celebrity face photos annotated with 40 binary attributes (e.g. smiling, mustache, brown hair, etc.). We consider a subset of 162,770 photos and solve the task of deciding, whether the image depicts a blonde person or not: this is the same setting considered in past works (Sagawa et al., 2020; Liu et al., 2021) that tackle the debiasing problem. The blonde/non blonde binary label is spuriously correlated with the gender label (the minority group blond-male contains only 1387 images).

BAR has been introduced in (Nam et al., 2020) as a realistic benchmark to test model's debiasing capabilities. It is constructed using several data sources and contains 1,941 photos of people performing several actions. The task is to distinguish between 6 categories: Climbing, Diving, Fishing, Racing, Throwing, and Vaulting. The bias arises from the context in which action photos are observed at training time: for instance, in training, diving actions are represented in a natural environment employing a diving suit, whereas in the test set, they are set in an artificial environment, e.g. a swimming pool (see Fig. 8 - right). For details, readers can refer to the original paper (Nam et al., 2020).

**Validation.** While previous methods (Nam et al., 2020; Liu et al., 2021; Levy et al., 2020), makes use of the bias annotations on the validation set for the above datasets for setting hyperparameters, we make a further effort in the direction of **unsupervised debiasing** and only validate the design choices and hyperparameters of our method for the synthetically corrupted CIFAR-10, as detailed later on in section 7. The same settings are then used for the realistic datasets without resorting to the validation set annotations for the bias, making our method fully unsupervised.

## 6.2 Performances

We report the performance of our approach on the different benchmarks above mentioned, all targeting a classification task. Accuracy is the metric adopted. We are primarily interested in improving accuracy on the unbiased test samples, those under-represented in the training data. However, at the same time, we do not want to decrease performance on the biased samples: in fact, a higher generalization implies that spurious correlations are not used anymore for classification, and this might cause a certain decrease of performance on some samples. For this reason, we report accuracies on the testing subset of unbiased samples, as well as over the entire test set (biased + unbiased).

**Implementation details.** All experiments comply the same evaluation protocol used in the competing methods for a fair comparison. We used ResNet-18 (He et al., 2016) as a backbone for Corrupted CIFAR-10 and BAR, and ResNet-50 as backbone for Waterbirds and CelebA. We remove the last layer from such backbones, adding a 2-layer MLP on top of it as a classifier head. As customary, ResNets are pre-trained on ImageNet (Krizhevsky et al., 2012b). The parameter network $h_\psi$ consists of a simple MLP with two hidden layers with 64 neurons each. We rely on the Pytorch implementation of the Beta distribution for sampling in a reparameterized fashion. We set the learning rate $\eta = 0.001$ for all datasets with batch size of 256 on synthetic biased data and 128 for realistic bias data. We used Adam (Kingma & Ba, 2015) as optimizer. In all experiments we adopted the prediction history (PH) method to infer the biased/unbiased samples. In our ablation analysis in Sec. 7, we discuss how different splitting strategies influence the final outcome, as well as further implementation details.

| Bias ratio | ERM | Supervised debiasing methods | | Unsupervised debiasing methods | | | |
|---|---|---|---|---|---|---|---|
| | | REPAIR Li & Vasconcelos (2019) | Group DRO Sagawa et al. (2020) | LfF Nam et al. (2020) | SelecMix Hwang et al. (2022) | BiaSwap Kim et al. (2021) | Ours |
| 95% | $45.2 \pm 0.22$ | 48.7 | 53.1 | 59.9 | 54.00 | 46.99 | **64.7 ± 1.20** |
| 98% | $30.2 \pm 0.77$ | 37.9 | 40.2 | 49.4 | 47.70 | 41.16 | **57.4 ± 1.15** |
| 99% | $22.7 \pm 0.97$ | 32.4 | 32.1 | 41.4 | 41.87 | 38.94 | **50.8 ± 1.03** |
| 99.5% | $17.9 \pm 0.86$ | 26.3 | 29.3 | 31.7 | 38.14 | 35.87 | **43.9 ± 0.87** |

Table 1: **Accuracy on the *whole* test samples of Corrupted CIFAR-10.** Accuracy (in %) evaluated on *biased + unbiased* test samples for different bias ratios. Results of REPAIR, Group DRO and LfF are from Nam et al. (2020). The performance of our proposed methods are obtained with $\omega = 10^{-3}$. Best performance in bold.

| Bias ratio | ERM | Supervised debiasing methods | | Unsupervised debiasing methods | | | |
|---|---|---|---|---|---|---|---|
| | | REPAIR Li & Vasconcelos (2019) | Group DRO Sagawa et al. (2020) | LfF Nam et al. (2020) | SelecMix Hwang et al. (2022) | BiaSwap Kim et al. (2021) | Ours |
| 95% | $39.4 \pm 0.75$ | 50.0 | 49.0 | 59.6 | 51.38 | 42.61 | **64.6 ± 1.14** |
| 98% | $22.6 \pm 0.45$ | 38.9 | 35.1 | 48.7 | 44.24 | 35.25 | **57.2 ± 1.09** |
| 99% | $14.2 \pm 0.91$ | 33.0 | 28.0 | 39.5 | 35.89 | 32.54 | **50.6 ± 1.01** |
| 99.5% | $10.5 \pm 0.28$ | 26.5 | 24.4 | 28.6 | 31.32 | 29.11 | **43.7 ± 0.99** |

Table 2: **Results on the *unbiased* test samples only of Corrupted CIFAR-10.** Accuracy (in %) evaluated *only* on the unbiased samples for different bias ratios. Results of REPAIR, Group-DRO and LfF are from Nam et al. (2020). The performance of our proposed methods are obtained with $\omega = 10^{-3}$. Best performance in bold.

**Performance for synthetic bias.** Tables 1 and 2 present the performances on synthetic biased datasets, reporting the average accuracy (mean + standard deviation) for the whole test set and for unbiased samples only, respectively. We compare against a model trained by Empirical Risk Minimization (ERM) as baseline, and different former debiasing methods, either using annotation for the bias or not. For the methods requiring explicit knowledge of the bias (i.e., supervised), we consider REPAIR (Li & Vasconcelos, 2019), which does sample upweighting, and Group DRO (Sagawa et al., 2020), which tackles the problem using robust optimization. We also compare our performance with that of unsupervised debiasing methods such

as Learning from Failure (LfF) (Nam et al., 2020), BiaSwap (Kim et al., 2021) and SelectMix (Hwang et al., 2022).

We consider different ratios of the bias (ranging from 95% up to 99.5%), as in (Nam et al., 2020), and many other former works. This ratio indicates the actual percentage of the dataset belonging to $\mathcal{D}_{bias}$ and $\mathcal{D}_{unbias}$, i.e., $|\mathcal{D}_{bias}|/(|\mathcal{D}_{unbias}| + |\mathcal{D}_{bias}|)$. The bias identification step has clearly an impact on the performance. We choose the PH method to estimate $\hat{\mathcal{D}}_{bias}$ and $\hat{\mathcal{D}}_{unbias}$ subsets, setting the hyperparameter $M = 5$ (number of epochs after which the ranking vector $\hat{s}$ is computed, see Sect. 4.2) for all experiments. This choice is consequence of an ablation study, showing the best performance on a validation set (see Fig. 9 and Sect. 7.1) We also set the hyperparameter $\omega$ weighting the regularization term to $\omega = 10^{-3}$ as a results of a careful ablation (Sec. 7.5).

For Corrupted CIFAR-10, our method always reaches the best accuracy. We can note here that the gap with former methods, both supervised and unsupervised, is much higher, ranging overall between about 5% and 20% regarding the whole test set (min and max difference across bias ratios, Table 1), and between about 5% and 22%, when testing on the biased samples only (Table 2). It is important to highlight that the results of our proposed method are reported as mean ± standard deviation, obtained after 5 run of the algorithms, while the accuracies reported by SelecMix and BiaSwap are the result of a single (presumably the best) run. Further, taking into account the standard deviation, our accuracies are comparable with the one of the winning methods.

Our results show that the obtained performances are rather stable and consistent across datasets and bias ratios, as compared to previous approaches.

| | | **ERM** | **Unsupervised debiasing** | | | | **Ours** | **Supervised** |
|---|---|---|---|---|---|---|---|---|
| | | | **CVar DRO** | **LfF** | **JTT** | **SSA** | | **Group DRO** |
| **Dataset** | **Test set** | | Levy et al. (2020) | Nam et al. (2020) | Liu et al. (2021) | Nam et al. (2022) | | Sagawa et al. (2020) |
| | | - | ✓ | ✓ | ✓ | 5% | × | ✓✓ |
| Waterbirds | *All* | 97.3 | **96.0** | 91.2 | 93.3 | 92.6 | 94.7 | 93.7 |
| | *Unbiased* | 72.6 *(24.7)*↕ | 75.9 *(20.1)*↕ | 78.0 *(13.2)*↕ | 86.7 *(6.6)*↕ | 87.1 *(5.5)*↕ | **87.7** *(7.0)*↕ | 91.4 *(2.3)*↕ |
| CelebA | *All* | 95.6 | 82.5 | 85.1 | 88.0 | **92.8** | 89.1 | 92.9 |
| | *Unbiased* | 47.2 *(48.4)*↕ | 64.4 *(18.1)*↕ | 77.2 *(7.9)*↕ | 81.1 *(6.9)*↕ | **86.7** *(6.1)*↕ | 82.2 *(6.9)*↕ | 88.9 *(4.0)*↕ |
| BAR | *All* | 53.5 | - | 62.9 | - | - | **64.8** | - |

Table 3: **Results on realistic datasets.** Accuracy (in %) evaluated on the test set composed by unbiased test samples (*Unbiased*) and the entire test set (*All*). The performance of our proposed methods are obtained with $\omega = 10^{-3}$. **Best** results among unsupervised debiasing methods for each dataset are in bold, and second-best are underlined. The gaps between *All* and *Unbiased* accuracy are also reported in brackets and indicated by ↕. The symbol ✓ refers to the use of validation set labels for the bias. SSA uses the full validation set and 5% of it, and it is also used in training. Conversely, our method doese not use any (×). Group DRO also uses training set labels for the bias (✓✓).

**Performance on the realistic biased datasets.** In these trials, we still compare against the ERM baseline, Group DRO as supervised debiasing method, and four unsupervised debiasing algorithms, LfF (Nam et al., 2020), CVaR DRO (Levy et al., 2020), JTT (Liu et al., 2021) and SSA (Nam et al., 2022) adopting the same protocol previously reported. Performances are shown in Table 3. For these datasets, we remind that we do not have the full control of the bias ratios. Specifically, in BAR we do not know exactly the biased/unbiased ratio and, differently from Corrupted CIFAR-10, which have a balanced test set, Waterbirds and CelebA test sets are also imbalanced. In these cases, as already highlighted previously, it is also important not only to cope with unbiased samples, but also to maintain accuracy on biased data. Consequently, it is paramount here to reach a good trade-off between generalizing to unbiased samples while keeping high performance on biased data as well. Hence, performances in Table 3 are reported as accuracies over both the entire test set (*All*) and the unbiased samples (*Unbiased*) for Waterbirds and CelebA, and over the whole test set only (*All*) for BAR.

For Waterbirds, we score favorably with respect to other unsupervised methods for the subset of *unbiased* samples: we outperform the best state-of-the-art (SSA (Nam et al., 2022)), despite they employ 5% of the validation set. We also surpass all other methods which make use of the full validation set. When addressing the entire test set (*All*) our method is the second best scoring -1.3% with respect to CVar DRO, followed by the other methods. In this respect, it is important to notice that CVar DRO, while it generally performs well on the entire, biased + unbiased test set, it drastically experiences large drops when considering the unbiased samples only, namely t20% and 18% for Watebirds and CelebA, respectively. The same behavior is evident for the standard ERM, with even larger gaps, due to the fact that there is no bias mitigation strategy in place.

For CelebA, we are the second best performers behind SSA (Nam et al., 2022) in both the full test set (*All*) and the *Unbiased* subset, altough the gap between the two remains the same (around 7%). We still score favorably against JTT (Liu et al., 2021) and surpass LfF (Nam et al., 2020) and CVar DRO (Levy et al., 2020) by a large margin.

Concerning the BAR dataset, since there is no ground-truth for the bias we report only the average accuracy over the whole test set: in this case, our method outperforms JTT (Liu et al., 2021) by a considerable margin.

Overall, our proposed method reaches a good trade-off, resulting competitive for both types of test sets, across the several datasets *without resorting to any bias group annotation on the validation set*. In other words, we are able to learn bias invariant representations without giving up accuracy on the biased samples, with a completely unknown bias. This is also analyzed in section 7.7, where we test on a supposedly unbiased dataset.

Finally, we show also competitive performance against the supervised method Group DRO: without using any bias supervision, on Waterbirds, our method surpasses its test accuracy on the entire test set, even if the accuracy on *biased* data only results lower (owing to the supervision in this case). On CelebA, the performance are less competitive with Group DRO, yet we still outperform other unsupervised debiasing methods.

## 7 Ablation analysis

We include here additional analyses and experiments which are useful to justify the methodological choices, while also presenting an extended ablation for the relevant hyperparameters. More in detail, we analyze (for convenience, the specific section is also indicated):

- **Sec. 7.1**: The bias identification approaches: we compare the subdivision obtained by Prediction History (PH) with the split obtained by Single Prediction (SP).

- **Sec. 7.2**: How the quality of the subdivision of the data in biased/unbiased samples affects the final accuracy.

- **Sec. 7.3**: How learning the Beta distribution parameters impacts the performance, namely, we compare our proposed *l-mix* strategy with the vanilla mixing mechanism *mixup*.

- **Sec. 7.4**: *vanilla mixup* performances by varying the parameter $\zeta$ controlling the contribution of the regularization term.

- **Sec. 7.5**: *l-mix* performances by varying the parameter $\omega$ controlling the contribution of the regularization term.

- **Sec. 7.6**: The possible augmentation policies (sampling strategies), and how these affect the performance at the different bias ratios.

- **Sec. 7.7**: An application of the proposed methods to an **unbiased** dataset showing that they are still beneficial in this scenario, i.e., generalization is improved with respect to standard ERM even with no apparent bias.

To have the full control of the experimental conditions, we conducted most of these trials and the ablation analysis using Corrupted CIFAR-10 (bias ratio= 95%) if not differently specified.

## 7.1 Quality of Bias Identification

We assessed the quality of the split obtained by the Prediction History (PH) method. We consider F1-score, Precision and Recall as metrics to evaluate the discrepancy between the pseudo-labels and the ground-truth bias/unbias split (see Fig. 9). CIFAR-10 data (bias ratio 95%) is considered. We evaluated the method over different values of $M$, namely the amount of epochs before refreshing the assignment: we can notice that as long as $M$ increases, recall decreases and precision increases, while F1-score is significantly higher than the random split case (see Table 4). We chose $M = 5$ for all our experiments, while different values are not affecting significantly the performance.

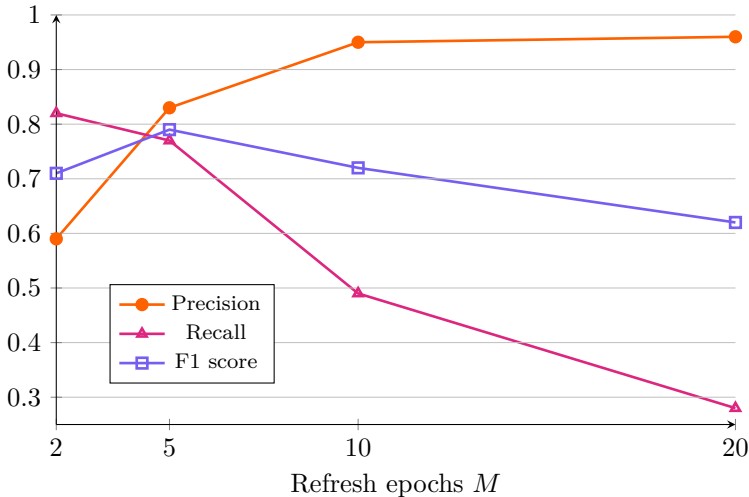

Figure 9: Performance of the Bias Identification method on Corrupted CIFAR-10 (bias ratio 0.95), for different parameter choice $M$, i.e. how often the ranking vector is updated (Eq. 3). We report Precision, Recall and F1-score as metrics.

## 7.2 Effect of Bias Identification on model performance

We investigate how different amounts of noise in the pseudo-labeling stage impact the final outcome in terms of test accuracy. We employed our proposed method on CIFAR-10, see Table 4. We show the test accuracies in the ideal case of perfect subdivision between biased and unbiased samples (oracle, $F1 = 1$), by applying our approach on top of Single Prediction and Prediction History procedures, and in the case of random split. We noted that passing from the oracle conditions (best) to the random split (worst), accuracy drops significantly. Interestingly, the method trained on the random pseudo-labeling always achieves better accuracy than that of the ERM baseline (see Table 1), and even higher than those achieved by some methods specifically designed to model debiasing. Even a coarse split (better than random) considerably increases the final performance with respect to ERM training. Also, our mixing strategy has a strong regularization effect even in suboptimal conditions (i.e., when mixing mostly biased samples). Comparing with the two bias identification strategies SP and PH, we see that the difference is not severe (drop between 1-4% for Single Prediction).

## 7.3 Learnable mix vs. vanilla mixup

In Table 5, we analyze how *learning* the parameters of the *Beta* distribution results in improved performance on both the whole test test (All) and the unbiased subset. The improvement is consistent across all datasets

| Bias ratio | Oracle | | Random split | | SP | | PH | |
|---|---|---|---|---|---|---|---|---|
| | F1 | Test Acc. | F1 | Test Acc. | F1 | Test Acc. | F1 | Test Acc. |
| 95% | 1.0 | 66.3 | 0.37 | 50.3 | 0.65 | 60.9 | 0.74 | **64.6** |
| 98% | 1.0 | 59.4 | 0.34 | 40.4 | 0.62 | 54.4 | 0.67 | **57.2** |
| 99% | 1.0 | 54.7 | 0.33 | 29.7 | 0.58 | 49.3 | 0.63 | **50.6** |
| 99.5% | 1.0 | 49.0 | 0.32 | 21.5 | 0.54 | 42.5 | 0.59 | **43.7** |

Table 4: **Ablation study on bias identification on Corrupted CIFAR-10.** The table reports F1 scores for $\hat{\mathcal{D}}_{bias}$ / $\hat{\mathcal{D}}_{unbias}$ splitting, obtained with our bias identification strategies Single Prediction (SP) and Prediction History (PH) presented in Section 4. We compare them to oracle (groundtruth) and randomly generated subsets. We report the final test accuracy on the whole test set for the four different cases. Best accuracies are obtained with oracle split, while the worst for random split. PH performs very favourably with respect to the oracle in terms of test accuracy, even if the split is not perfect.

analyzed. We set $\zeta = 10$ for the vanilla mixup (Zhang et al., 2018) strategy and $\omega = 10^{-3}$ for our proposed learnable strategy *l-mix* (for the latter, also check the ablation study in Sect. 7.5).

| | CIFAR-10 (95%) | | Waterbirds | | CelebA | | BAR |
|---|---|---|---|---|---|---|---|
| | All | Unbiased | All | Unbiased | All | Unbiased | All |
| *vanilla mixup* | 63.3 | 63.3 | 94.3 | 87.1 | 88.3 | 81.6 | 64.3 |
| *learnable mix* | **64.7** | **64.6** | **94.7** | **87.7** | **89.1** | **82.1** | **64.8** |

Table 5: Ablation study on vanilla mixup vs. learnable strategy *l-mix*. For the latter, the parameters $\alpha$ and $\beta$ of the Beta distribution are learned by the neural network $h_\psi$ (equations 7 and 9). For vanilla mixup, the parameters are kept fixed (Zhang et al., 2018). In our experiments, we set $\zeta = 10$ and $\omega = 10^{-3}$.

### 7.4 Ablation study on $\zeta$ hyperparameter for vanilla mixup

We also conducted an analysis on the impact of the $\zeta$ hyperparameter acting as regularization term in the loss $\mathcal{L}_{vanilla}$ (Eq. 6) adopting the vanilla mixup technique. We investigated $\zeta$ in the range $[0, 10^0, 10^1, 10^2, 10^3]$ and evaluated the performance on Corrupted CIFAR-10 on both biased and unbiased test samples, see Fig. 10. Even in this case, we observe a large range in which the method is benefiting from the regularization effect, showing accuracies always better than the case of $\zeta = 0$ representing the standard ERM training. As reported in Sect. 6, we always used $\zeta = 10$ for all our experiments.

### 7.5 Ablation study on $\omega$ hyperparameter

Similarly, we set different values of the parameter $\omega$ of Eq. 11, i.e., $[0, 10^{-4}, 10^{-3}, 10^{-2}, 10^{-1}]$, and evaluated the performance of our learnable mechanism *l-mix* on Corrupted CIFAR-10 (bias ratio = 95%). In Table 6, we report the accuracy on the unbiased and on the full test set. We can note that there is a large range of $\omega$ values, $10^{-4} < \omega < 10^{-2}$, in which the accuracy is reaching higher values with respect to the the case $\omega = 0$ (please, note the logarithmic scale in the $x$ axis). This empirically shows that $\omega$ is not a so sensitive parameter, and for this reason, we fix $\omega = 10^{-3}$ in all our experiments.

### 7.6 Augmentation strategies

We report in Table 7 an ablation study on the augmentation strategies, consisting in sampling $\hat{x}_1$, $\hat{x}_2$ in different ways from either $\hat{\mathcal{D}}_{bias}$ or $\hat{\mathcal{D}}_{unbias}$ or both. We consider the Corrupted CIFAR-10 benchmark, and show the final accuracies for the unbiased and full test sets, for all the bias ratios considered.
From the figures on the table, one can note that sampling both $\hat{x}_1$, $\hat{x}_2$ from $\hat{\mathcal{D}}_{bias}$ overfits the biased data and results in the worst accuracy, while mixing both samples from $\hat{\mathcal{D}}_{unbias}$ increases the generalization over

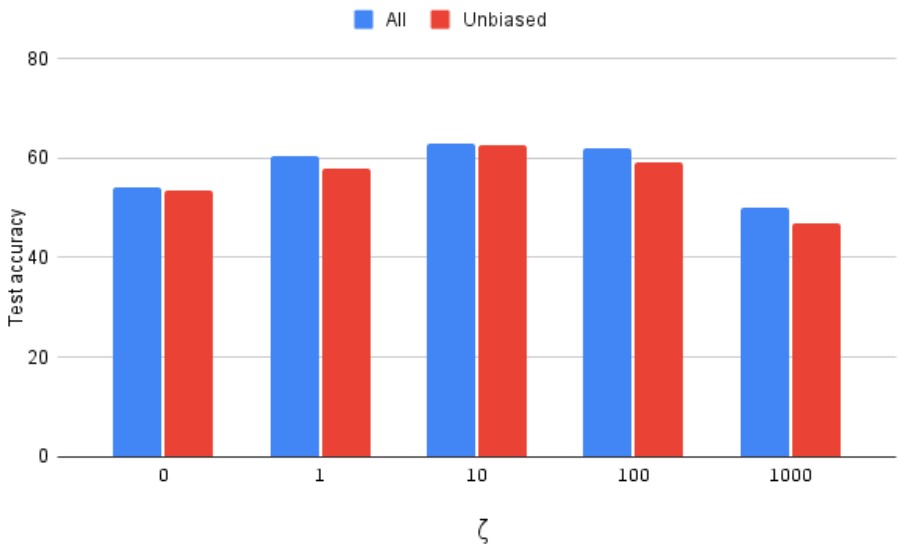

Figure 10: Test accuracy for unbiased samples (red) and full set of samples (blue) for different values of the hyperparameter $\zeta$. For $\zeta = 0.0$, the loss corresponds only to the weighted ERM of Eq. 6. Note that $\zeta$ axis is in logarithmic scale.

| $\omega$ | Corrupted CIFAR-10 | | Waterbirds | | CelebA | |
|---|---|---|---|---|---|---|
| | All | Unbiased | All | Unbiased | All | Unbiased |
| 0 | 61.8 | 62.0 | 91.6 | 84.3 | 85.4 | 78.9 |
| $10^{-4}$ | 63.7 | 63.8 | 92.6 | 86.2 | 87.1 | 80.6 |
| $10^{-3}$ | **64.7** | **64.6** | **94.7** | **87.7** | **89.1** | **82.2** |
| $10^{-2}$ | 61.9 | 62.2 | 91.9 | 85.1 | 86.7 | 80.1 |
| $10^{-1}$ | 60.6 | 61.5 | 89.6 | 82.3 | 83.8 | 75.7 |

Table 6: Test accuracy for unbiased samples and the full test set (All) for different values of the hyperparameter $\omega$ evaluated on different benchmarks. The values for $\omega$ are chosen on a logarithmic scale.

unbiased samples, but provides suboptimal results, especially for the biased test set. Mixing samples from $\hat{\mathcal{D}}_{bias}$ and $\hat{\mathcal{D}}_{unbias}$, corresponding to our strategy *l-mix*, provides the best accuracy for all bias ratios.

In order to decouple the sampling strategy from the learning problem, we also considered the vanilla Mixup data-augmentation as a baseline (2nd and 7th rows in Table 7). Results are slightly higher than those of mixing $\hat{\mathcal{D}}_{bias}$ - $\hat{\mathcal{D}}_{bias}$ (3rd and 8th rows), but lower than the other combinations. This was somehow expected as most of the samples in the dataset are biased, therefore it is very likely that, by selecting pairs to be mixed randomly, biased/biased pairs are picked more often, while biased/unbiased or unbiased/unbiased pairs are less frequently considered. We also report the baseline case in which no augmentation is performed (1st and 6th rows), i.e. $x_{mix}$, $y_{mix}$ in equations 5 and 6 are just individual samples randomly drawn from $\mathcal{D}$: performances in this case are significantly lower than those obtained by our proposed approach.

To conclude, this analysis proves that by choosing pairs from $\hat{\mathcal{D}}_{bias}$ and $\hat{\mathcal{D}}_{unbias}$ is much more effective, even when the two splits are not accurate.

## 7.7 Impact on unbiased datasets

To conclude, we tested our proposed methods on the standard CIFAR-10 dataset, without any induced bias. With this test, we

| | Test accuracy | |
|---|---|---|
| ERM | *vanilla mixup* | *l-mix* |
| 92.94 | 93.65 | **94.02** |

Table 8: Ablation study on unbiased

| Set 1 | Set 2 | Bias ratio | | | | |
|---|---|---|---|---|---|---|
| | | 95% | 98% | 99% | 99.5% | |
| No augmentation | | 58.8% | 46.1% | 40.0% | 33.6% | |
| Vanilla MixUp | | 44.2% | 39.4% | 37.2% | 35.2% | All |
| $\hat{\mathcal{D}}_{bias}$ | $\hat{\mathcal{D}}_{bias}$ | 35.2% | 34.0% | 32.9% | 32.0% | |
| $\hat{\mathcal{D}}_{unbias}$ | $\hat{\mathcal{D}}_{unbias}$ | 60.2% | 54.1% | 48.4% | 40.4% | |
| $\hat{\mathcal{D}}_{bias}$ | $\hat{\mathcal{D}}_{unbias}$ | **63.8%** | **56.4%** | **50.9%** | **43.1%** | |
| No augmentation | | 55.3% | 41.5% | 34.8% | 27.1% | Unbiased Only |
| Vanilla MixUp | | 42.1% | 36.7% | 36.5% | 34.4% | |
| $\hat{\mathcal{D}}_{bias}$ | $\hat{\mathcal{D}}_{bias}$ | 29.7% | 28.4% | 26.7% | 27.5% | |
| $\hat{\mathcal{D}}_{unbias}$ | $\hat{\mathcal{D}}_{unbias}$ | 63.1% | 55.3% | 48.7% | 42.5% | |
| $\hat{\mathcal{D}}_{bias}$ | $\hat{\mathcal{D}}_{unbias}$ | **63.3%** | **55.9%** | **49.4%** | **42.7%** | |

Table 7: **Ablation analysis on the augmentation strategies.** For Corrupted CIFAR-10, we report the accuracy resulting from different augmentation strategies and no augmentation, by varying the bias ratio. Our strategy results the winner over all the other mixing policies for both the entire test set (*All*) and the *Unbiased Only* portion (best results are in bold).

want to prove that even in the presence of a non-biased dataset, our proposed method is beneficial, showing an improved generalization capacity. In other words, we can tackle a real-world unsupervised debiasing problem, in which we do not actually know a priori whether a dataset is biased or not. We expect that our proposed method does not underperform with respect to standard ERM training.

Differently from Corrupted CIFAR-10, the standard version of the dataset does not exhibit any known shortcut when it comes to predict the class labels. We compared our proposed method *l-mix*, the non-learned version, vanilla mixup, and standard ERM, using the same architecture $f_\theta$ (ResNet-18) as backbone for a fair comparison. The dataset split is performed with the PH approach ($M = 5$), and regularization parameters were set to $\zeta = 10$ for vanilla mixup, and $\omega = 10^{-3}$ for *l-mix*. We can observe that in both cases, our results exceed the ERM baseline, thus showing that our approaches are agnostic to the fact that the dataset might be biased or not, making them more amenable to be utilized in real use cases.

## 8    Conclusions

In this work, we address the problem of data bias in the realistic, unsupervised debiasing scenario, i.e., when the bias factor or attribute is unknown, and propose a novel approach to mitigate this issue. Our proposed method is composed of two sequential stages. First, we introduce two strategies to subdivide the training dataset into two subsets (biased and unbiased samples), namely SP and PH. Relying on the history of predictions, PH results in cleaner separation of biased/unbiased samples, which in turn results in better performance for the subsequent bias mitigation stage.

Second, we show how to effectively exploit such subdivision in order to produce augmented samples by mixing estimated biased and unbiased samples by properly learning mixing coefficients in an adversarial way. Such mixed samples act as a regularizer which breaks the spurious correlations between data and class labels, thus increasing accuracy.

The overall approach scores favorably against state of the art approaches and is also quite robust to the choice of hyper-parameters, for which we present a comprehensive ablation analysis. Notably, the method

is completely agnostic of the presence of bias, and it also outperforms standard ERM even when no bias is (apparently) present in the data.

**Broader Impact Statement**

This research addresses the challenge of developing AI systems that can learn effectively from data affected by bias, especially in the realistic case of unknown bias. The goal is then to mitigate the bias effect in training that could lead to limited generalization and unfair or harmful outcomes. This is particularly important in all actual scenarios, i.e., whenever one does not know *a priori* whether the real dataset collected evidences some form of bias or not. Our system proved to perform well in both cases, being able to reach top performances even when the dataset is unbiased, demonstrating that its mechanism is adaptative to the different data conditions. More in general, effective generalization from biased data can enable AI systems to perform well across diverse populations, improving fairness and inclusivity in their applications. By introducing methods to detect and mitigate bias during learning, this work contributes to the development of more equitable and transparent AI systems. Nonetheless, careful evaluation and ethical oversight are essential to ensure these methods are applied responsibly, particularly in real-world, high-impact settings.

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

## A    Appendix

You may include other additional sections here.

