# OpenReview forum: "Model Debiasing by Learnable Data Augmentation"
_TMLR — Rejected by TMLR_

### Review · Reviewer_LxSz · 2025-03-10

**Summary Of Contributions:**

This paper addresses the important problem of neural network models learning biased representations due to spurious correlations in training data. It proposes a novel two-stage approach for unsupervised debiasing - when the bias is unknown.
Its contributions are as follows:
1. A method to identify biased/unbiased samples by observing model training behavior
2. A learnable data augmentation strategy that mixes biased and unbiased samples to regularize training
3. Strong empirical results on synthetic and realistic biased datasets, outperforming previous methods

**Audience:**

Yes

**Claims And Evidence:**

Yes

**Requested Changes:**

Please refer to the weakness.

**Strengths And Weaknesses:**

Strengths:
1. The paper tackles the realistic unsupervised debiasing scenario, where bias annotations aren't available - a common real-world situation.
2. The learnable data augmentation strategy (l-mix) is innovative, allowing the model to adaptively determine optimal mixing parameters rather than using fixed values.
3. The method consistently outperforms existing techniques across various datasets and bias ratios, showing 5-22% improvements in some cases.
4. Testing spans synthetic datasets with controlled bias (Corrupted CIFAR-10) and realistic biased datasets (Waterbirds, CelebA, BAR).
Robust Ablation Study: The paper includes a thorough analysis of all components and hyperparameters.
5. The approach improves performance even on seemingly unbiased datasets, showing it enhances general model regularization.

Weaknesses:
1. The method requires multiple forward passes through the entire training set (e.g., in Identification by Prediction History and Learnable Data Augmentation), which could be computationally expensive.
2. The paper fails to properly justify why data augmentation was chosen, why it was limited to Mixup, and why improving Mixup specifically was necessary.
3. The paper lacks an overall framework diagram that would help readers better understand the proposed approach.
4. In the related work section, references to causal inference are missing, despite "shortcuts" and spurious correlations being closely related to this field.
5. There's no explanation for why equation 4 and equation 2 use the same hyperparameter γ.
6. The paper only considers cross-mixing between biased and unbiased sets without exploring mixing within the same sets.
7. There are numerous typos in the paper, such as: In the first paragraph of the introduction: "reaHching" instead of "reaching", in the fourth paragraph of the introduction: "To this end, We" (capitalized W after a comma), and in Figure 2: issues with the "Rho" label.
8. While empirically effective, the paper lacks a theoretical analysis of why learning mixing parameters works better than fixed parameters.

---

> ### Author Response · Authors · 2025-03-28
>
> Thank you very much for the useful feedback, which helped us to improve the manuscript. Here follows a discussion on the raised points. The manuscript has been changed to include further discussion when needed  (changed or added text is in red for your convenience).
>
> 1. The method requires two training stages, similarly to JTT (Just Train Twice), Liu et al. (2021) and SSA, Nam et al. (2022). The first stage requires training a model with ERM with no additional forward passes (the standard training iterations are enough to collect the prediction history); the second stage has a more complex objective, as formulated in Eq. 11, which does not introduce significant overhead.
>
> 2. Data augmentation is chosen to attain oversampling of unbiased samples, which is a strategy sometimes used in bias removal methods. Other methods used resampling/reweighting, which has the same goal of data augmentation. Mixup is a state-of-the-art augmentation tool, which also allows us to incorporate unbiased data and produce augmentations which are more effective for model debiasing. This has been clarified in section 5.
>
> 3. Thanks for the suggestion, we have included an initial extra figure to better convey the idea of the overall two-stage pipeline (Figure 2).
>
> 4. Thanks for the suggestion. We agree that this is a relevant affine theme and we added a paragraph in the Related Work section (Sect. 2), citing two papers that we deem interesting in the context of our work.
>
> 5. Yes, equation 4 and 2 share the same hyperparameter $\gamma$. As explained in section 4.1, this is exactly the fraction of samples assigned to $\hat{\mathcal{D}}\_{bias}$  (thus $1-\gamma$ is the fraction of samples assigned to $\hat{\mathcal{D}}_{unbias}$). Therefore, as explained right below Equation 4, it makes sense to weigh the contribution of biased and unbiased samples with coefficients $1-\gamma$ and $\gamma$, respectively, in order to deal with data imbalance. This has been further clarified in the text.
>
> 6. The paper already contained this information: Table 7 ablates cross-mixing with all different combinations, together with vanilla mixup (namely, random mixing of samples and related labels). The best combination resulted in the one that mixes biased and unbiased samples. Even if not intuitive, vanilla mixup is overall detrimental since the high frequency of biased (bias-aligned) samples makes the mixing within $\hat{\mathcal{D}}_{bias}$ extremely likely (namely, mostly biased samples are mixed together), while the mixing of unbiased samples between them or with biased ones will result less probable. Such augmentations have the effect of amplifying the bias, as demonstrated quantitatively in Table 7.
>
> 7. Thank you for the remark, we have fixed all typos. Figure 2 (now Figure 3) was also improved as suggested by reviewer qXQf.
>
> 8. In learning the mixing parameters, we followed the common intuition that learning them, i.e., optimizing them in a data-driven fashion, usually works better than using a fixed (random) combination, as in the vanilla mixup. Our empirical results are obtained by following a formal, technical sound procedure corroborating the intuition.

---

### Review · Reviewer_qXQf · 2025-03-11

**Summary Of Contributions:**

This paper considers the problem of bias mitigation on image classification tasks. The proposed solution is a two-stage strategy: (1) Identifying bias-aligned samples from the training dataset using the (history of) classification result after a full vanilla training. (2) Mixup of bias-aligned and bias-conflicting samples with a jointly optimized hyper-parameters of the mixture-ratio-generating distribution. The trained model achieves higher average and worst-group accuracies than classic baselines (e.g., LfF, JTT) on the corrupted-CIFAR-10, Waterbirds, and CelebA.

**Audience:**

Yes

**Broader Impact Concerns:**

Broader impact section is already there and does not need further refinement.

**Claims And Evidence:**

No

**Requested Changes:**

For details, see the "weaknesses" above.
- Add a paragraph to clarify the use of validation sets and HP tuning.
- Add baselines to clarify the value of bias-annotation-based mixup.
- Add more recent baselines, such as SSA.
- Address minor clarity issues.

**Strengths And Weaknesses:**

### **Strengths**
- **Soundness of the approach.** The idea to mixup the bias-aligned and bias-conflicting samples makes much sense. While I do not believe that the emergence of bias should not be fully attributed to the matter of "dataset size" (as observations of Nam et al. (2020) suggests), mixup is definitely a sensible way to resolve any issue that has been caused by the cardinality imbalance.
- **Clarity.** The paper is written with a satisfactory clarity, except for some minor issues.
- **Technicality.** For the optimization of $\alpha,\beta$, the paper develops a nice adaptation of existing frameworks.

### **Weaknesses**
- **Limited to image classification.** A clear limitation of the proposed framework is that its direct application is limited to debiasing image classification models. This is because the method critically relies on two tools, (1) bias set identification based on correct predictions, and (2) mixup.
- **Need more decoupling with gains from Mixup.** One possibility that I am worried about is that all the claimed benefit comes from the performance gain provided by Mixup, orthogonal to the matter of bias. In fact, adding any good augmentation to existing methods is likely to boost the performance of any debiasing algorithm, in average and in worst-group accuracies. If my understanding is correct, the key claim of the paper is that we should perform mixup across bias-aligned and bias-conflicting samples, not mixing samples with different labels (which is what original mixup would do). To make this point concrete, I think it is necessary to compare against following additional baselines. (1) The proposed method with label-based mixup. (2) JTT and LfF baselines with label-based mixups.
- **Not quite clear on "unsupervisedness."** After reading the paper, it is not 100% clear to me that the use of bias annotations has remained at the level where we can call this method unsupervised in a strict sense. Indeed, the paper relies on many empirical observations based on bias annotations (which are also used for evaluation), and also utilizes multiple hyperparameters $\gamma, \zeta, \omega \ldots$. Regarding the hyperparameters, I appreciate the fact that the paper makes much effort to remove the need for an extensive tuning of the hyperparameters, such as using a fixed choice of hyperparameters throughout the paper. However, it is not very clear to me how $\gamma$ has been decided and/or may not require validation-set-based tuning on large-scale datasets where achieving high training accuracy may be tough. It would be great if authors could add a summary paragraph or section devoted for clarifying the unsupervisedness of the method, including discussions on the use of validation samples and tuning of all hyperparameters.
- **Baselines.** I wonder how the performance of the proposed model will compare with more recent baselines. One method that comes to my mind is "spread spurious attribute (SSA)" by the authors of LfF.
- **Minor clarity issues.** There is an unfinished sentence in the first paragraph of page 10 (the last sentence); the image quality of fig 2 is quite bad; it is unclear why Fig.4 only contains triangle samples, not ellipses.

---

> ### Author Response · Authors · 2025-03-28
>
> Thank you very much for your comments and constructive feedback.
> As per the limitation of our approach to image classification, we do not actually consider it as a limitation. In fact, we respectfully notice that a large part of the literature addressing bias mitigation involves image classification only, since it is one of the more classic tasks that is affected by the data bias problem.
>
> As per the 4 points raised (“Requested Changes”), which are also directly referring to the comments in “Weaknesses”:
>
> > Add a paragraph to clarify the use of validation sets and HP tuning.
>
> We have added a paragraph in section 6.1 (text is in red for your convenience): while previous methods (LfF, JTT, CVaR_DRO), make use of the bias annotations on the validation sets, we make a further effort in the direction of unsupervised debiasing and only validate the design choices and hyperparameters of our method for the synthetically corrupted CIFAR-10, as detailed later on in section 7. The same settings are then used for the realistic datasets without resorting to any validation set annotations for the bias, making our method fully unsupervised. This is also further expanded in Table 3 (and related discussion), where we clarify that previous methods make use of validation sets.
>
> _Concerning the validation of $\gamma$:_
>
> $\gamma$ was fixed to 0.85, considered as the accuracy of the trained model on the training set itself, i.e., the percentage of training samples correctly classified with respect to the entire training set. Actually, this value is chosen as a reasonable accuracy value, easily achievable when a model, ERM-trained on a biased dataset, is tested on the same training data. In fact, biased datasets adopted in literature have a bias ratio between 95% and 99.5%, so fixing $\gamma$ to 0.85 is surely an achievable accuracy in training, being the model mostly overfitting biased data.
>
> This value was fixed once, after exploration on CIFAR-10, and maintained on all experiments for all datasets. Different values for $\gamma$ have not been shown to provide significant discrepancies in the final performance of the bias-mitigated models.
> The discussion on  $\gamma$ was expanded for better clarity after equation (2) and equation (4).
>
> >  If my understanding is correct, the key claim of the paper is that we should perform mixup across bias-aligned and bias-conflicting samples, not mixing samples with different labels
>
> Yes, that is exactly the claim. However, in doing so, we pick two random samples (thus with random labels) and perform label mixup as well, as reported in equation 5.
>
> > Add baselines to clarify the value of bias-annotation-based mixup.
>
> The suggested baselines are already included in Section 7.6 - Table 7, where we ablate different mixing strategies. Even if not intuitive, vanilla mixup is overall detrimental since the high frequency of biased (bias-aligned) samples makes the mixing within $\mathcal{D}_{bias}$ extremely likely (namely, mostly biased samples are mixed together), while the mixing of unbiased samples between them or with biased ones will result less likely. Such augmentations have the effect of amplifying the bias, as demonstrated quantitatively in Table 7. For this reason, combining vanilla mixup with JTT e LfF would also be detrimental. In fact, as shown in most previous works (including JTT and LfF), the increase of the bias ratio in the data always results in worse performance since it is more difficult to cope with the smaller number of unbiased samples and give them the right importance for the sake of bias mitigation.
>
> > Add more recent baselines, such as SSA.
>
> We have added the results for SSA in Table 3 and discussed the method in the Related Works section. As other methods like LfF, JTT, and CVaR_DRO, SSA makes use of a validation set (either the full set or 5% of it) with bias annotations in order to tune the hyperparameters. More specifically, SSA follows a 2-stage strategy, in which the 1st stage consists in pseudo-labeling the samples as biased or unbiased as we also do, but with the help of a group-labeled set (validation samples with bias attribute annotations), which is not used in our approach. SSA in fact makes strong use of this labeled data in both training (during pseudo-labeling) and model validation (2nd stage). Despite this, our proposed method, without any group-labeled annotation, is reaching better performance on Waterbirds.
>
> > Address minor clarity issues.
>
> Thank you, we have improved the quality of Figure 3 (former Figure 2) and Figure 9 (former Figure 8), which had similar quality. Figure 5 (former Figure 4) has been modified to also include ellipses, which were indeed missing. The unfinished sentence has been fixed.

---

> ### Author Response · Authors · 2025-03-28
> **More details on hyper-parameter validation**
>
> Concerning the validation of the other hyperparameters, namely, $\zeta$ and $\omega$, we discussed both in the ablation analysis of Sect. 7, precisely, Sect. 7.4 and 7.5, respectively.
>
> $\zeta$ is the hyperparameter modulating the loss term related to the mixed samples added to the weighted ERM loss terms in Eq. 6. The benefit of such regularizer is analyzed on Corrupted CIFAR-10 (bias ratio 95%), showing a stable accuracy increase over the standard (weighted) ERM ($\zeta = 0$), as well as little sensitivity (see Fig. 9 in Sect. 7.4 - note that the x-axis is in log scale). $\zeta$ was fixed to the value of 10 for all other experiments.
>
> $\omega$ modulates the regularization term $\textit{Reg}$ in the final loss $\mathcal{L}_{debias}$, reported in Eq. 11. Similarly to $\zeta$, $\omega$ is ablated over Corrupted CIFAR-10 (bias ratio 95%): also in this case, we notice a stable improvement with respect to the case $\omega = 0$ and a little accuracy sensitivity over a large range of values, from $10^{-4}$ to $10^{-2}$. $\omega$ was fixed to the value of $10^{-3}$ for the other experiments.
>
> Indeed, all hyperparameters in play were actually fixed with respect to the synthetic dataset Corrupted CIFAR-10 and not ablated on the other datasets, thus making our method really unsupervised.

---

### Review · Reviewer_jPKv · 2025-03-14

**Summary Of Contributions:**

This paper aims to address spurious correlations between domain information and class labels caused by domain shifts from datasets, without available domain labels. The authors introduce a framework that identifies minority groups within the data based on the model’s fitting to training samples. Then, inter-group data augmentation and re-weight are performed to learn representations that mitigate spurious correlations. Extensive experiments demonstrate that the proposed framework consistently achieves stable improvements on both synthetic datasets and real-world datasets with domain shifts.

**Audience:**

Yes

**Broader Impact Concerns:**

There are no broader impact concerns for this paper.

**Claims And Evidence:**

Yes

**Requested Changes:**

More appropriate terminology or a further clarification of the meanings of “bias/unbias” would be beneficial.

**Strengths And Weaknesses:**

Strengths:
- The paper is well-written and easy to follow.
- The paper is well-motivated, with detailed and valid derivations and explanatory experiments.
- Thorough experiments confirm the effectiveness of the proposed approach in improving both minority-group performance and overall accuracy. Detailed ablation studies further validate the contribution of each proposed component.

Weaknesses:

I believe this paper is well-written, with conclusions clearly explained and validated. There are no significant shortcomings. The only minor issue is related to terminology:
- Ambiguity in the usage of “bias/unbias”: The paper uses “bias” and “unbias” to refer to majority and minority groups i.e. $D_{bias}$ and $D_{unbias}$, respectively. The term “bias” may imply that data from these groups is of lower quality or detrimental to training, while the actual difference is only in their frequency. I suggest adopting the more standard terms “majority” and “minority,” or "well-represented" and "under-represented", to avoid this ambiguity.

---

> ### Author Response · Authors · 2025-03-28
>
> Thank you very much for the positive feedback on our work.
>
> Concerning the remark on the terminology used for $\mathcal{D}\_{bias}$  and $\mathcal{D}\_{unbias}$
>
> > More appropriate terminology or a further clarification of the meanings of “bias/unbias” would be beneficial.
>
> Yes, we use the terms “bias” and “unbias” to refer to majority and minority groups, as correctly noted, with no implications on the quality of the samples. This is common in the literature, which sometimes refers to the two subset as 'bias-conflicting' and 'bias-aligned' as in Nam et al. (2020). We prefer to keep the terminology more compact and use:
> - $\mathcal{D}_{bias}$ for the majority group
> - $\mathcal{D}_{unbias}$ for the minority group.
>
> This has been clarified in the second paragraph of section 3, following your suggestion (text is in red for your convenience).

---

### Decision · Action_Editor_T2wg · 2025-06-13

**Recommendation:** Reject

**Comment:**

The paper needs to strengthen the claims regarding the choice of augmentation strategy. This would not be a minor revision and would require experimentation to validate. Hence, this would be classed as a major revision. Based on this, I am recommending a major revision for this work.

**Audience:**

The work would be of interest as there are people interested in model debiasing.

**Claims And Evidence:**

The key claim of the paper is that we should perform mixup across bias-aligned and bias-conflicting samples, not mixing samples with different labels. Two reviewers agree that the claims are validated. One of the reviewer is not convinced. The reviewer who is not convinced has two main points. One is the need for two passes over the data. As the method aims to use a bias identification and augmentation using mixup, it is hard to see how the two passes over the data can be avoided. The second point is the use of mixup and not comparing other augmentation strategies. This, I believe is a valid point. The authors agree that there are other strategies like data resampling/reweighting. They maintain that mixup is state-of-the-art method for augmentation. While mixup is a valid method for data augmentation, the point that is not validated is whether the choice of augmentation method (i.e. mixup) vs others is crucial for the task of debiasing. Would other augmentation strategies have also resulted in valid debiasing or not. They would have to follow the procedure of not mixing samples with different labels. This point of the claim I believe is not validated and does need validation.

**Resubmission Of Major Revision:**

The authors may consider submitting a major revision at a later time.

---

> ### Author Response · Authors · 2025-07-03
> **Official Comment by Authors - part 1**
>
> Dear Action Editor,
>
> Thanks for taking the time to process our submission.
>
> We would respectfully like to discuss with you and raise a few points about the review process as it was conducted for our paper, and the criticisms we received.
>
> As per the review process, we were expecting a different procedure, as also quoted by the letter from the Editor-in-Chief received after the first review phase, which stated that “… *To maximise the period of interaction and discussion, please respond as soon as possible. The reviewers will be using this time period to hear from you and gather all the information they need. In about 2 weeks, and no later than 4 weeks, reviewers will submit their formal decision recommendation to the Action Editor in charge of your submission.*”
>
> We promptly discussed the issues raised by the reviewers and we were expecting that the reviewers would finalize their scores after a phase of interaction with us about the remarks, unclear points or misunderstandings, and that it would have been an opportunity for discussion, before coming up to the final recommendations. Indeed, we expected a post-rebuttal phase involving updated reviews or ratings preceded by discussions and clarifications. This is also confirmed by the experience of one of the authors who acted as TMLR reviewer.
>
> This was not the case. Unfortunately, after our reply to the reviewers’ concerns, we did not receive any request of clarifications nor we noticed finalized reviews, and we finally received just the notification of the final decision by the Action Editor. Indeed, we think that, instead, a discussion with Reviewers may have helped a lot in the evaluation of our work and support the AE in his decision.
>
> Regarding the decision “*Resubmission of Major Revision: The authors may consider submitting a major revision at a later time*”, it is unclear whether this implies a resubmission to the same Reviewers as a revision of the current work. In fact, the requested experiments with standard augmentation can be easily included as a baseline in the manuscript if needed.

---

> ### Author Response · Authors · 2025-07-03
> **Official Comment by Authors - part 2**
>
> In this respect, however, we would also like to respond to the AE’s conclusions, particularly regarding the use of alternative augmentation strategies, which seems to be a major critical point.
>
> Our mixup-based augmentation allows the mixing of biased and unbiased samples (by learning how to weight the mix), regardless of the semantic task labels. Standard augmentations (e.g., color jitter, cropping, scaling) **do not** fit with our approach and cannot be compared with it. Specifically, comparing the two is not meaningful simply because standard augmentations are not designed for debiasing. Other augmentation methods cannot be integrated into our pipeline and are not beneficial for debiasing purposes. In fact, standard augmentations can help preventing overfitting in general, but they do so regardless of bias status of the samples. We can of course provide results using standard augmentations by considering indifferently all samples (i.e., regardless of their status of bias-aligned or bias-conflict), but we do not think this could provide useful insights. We instead provided in Table 7 an ablation of all possible mixing strategies (see below).
>
> As for the comment on reweighing and resampling — “*The authors agree that there are other strategies like data resampling/reweighing*” — we would like to emphasize that these are not augmentation strategies but, rather, alternative debiasing techniques, which we also address in the manuscript (e.g. weighted ERM).
>
> Finally, in response to the comment “*They would have to follow the procedure of not mixing samples with different labels*”, we would like to clarify that a misunderstanding here may have been generated by the different meaning of ‘labels’. Labels refer to downstream tasks labels (classification), while pseudo-labels refer to bias/unbias estimation (carried out in step 1). Our method mixes samples regardless of tasks labels: indeed, we randomly pick samples from the estimated unbiased and biased subset and mix them, along with their task labels, as per eq. 5. What matters is mixing samples across bias/unbias splits of the dataset, i.e. samples with different “bias” pseudo-labels. In this context, we considered in Table 7 all possible combination cases as an ablation study, where we mix samples with the same “bias” pseudo-labels, , specifically, we report results for no-augmentation, vanilla mixup, and our proposed mixing strategies sampling data from the sets of estimated biased and unbiased sets for all combinations.
>
> Concerning the comment about the 2 forward passes, we would like to raise the point that in the debiasing literature there are a lot of methods adopting our same strategy or employing an auxiliary model which accounts for a second forward pass. For instance Just Train Twice (Liu et al., 2021) trains the model twice, Learning from Failure (Nam et al., 2020) uses an auxiliary model, and other works (Teney et al., 2022;  Li et al., 2022; Kim et al., 2022a; Liu et al., 2021; Lemoine et al., 2018) are in the same line, as detailed in Section 2. This is due because the bias is unknown. So, we do not see this as a weak aspect of our work.
>
> In conclusion, considering the above comments, we would be grateful if you could reconsider your decision, also given the fact the two reviewers out of three were positive about our paper. We remain available for any other clarification or further information you may need.